# Iterative Teacher-Aware Learning

**Luyao Yuan**[1]
yuanluyao@ucla.edu

**Dongruo Zhou**[1]
drzhou@cs.ucla.edu

**Juhong Shen**[2]
jhshen@ucla.edu

**Jingdong Gao**[1]
mxuan@ucla.edu

**Jeffrey L. Chen**[1]
jlchen0@ucla.edu

**Quanquan Gu**[1]
qgu@cs.ucla.edu

**Ying Nian Wu**[3]
ywu@stat.ucla.edu

**Song-Chun Zhu**[1,3,4]
sczhu@stat.ucla.edu

[1]Department of Computer Science, [2]Department of Mathematics, [3]Department of Statistics
University of California, Los Angeles
[4]Beijing Institute for General Artificial Intelligence (BIGAI)

## Abstract

In human pedagogy, teachers and students can interact adaptively to maximize communication efficiency. The teacher adjusts her teaching method for different students, and the student, after getting familiar with the teacher's instruction mechanism, can infer the teacher's intention to learn faster. Recently, the benefits of integrating this cooperative pedagogy into machine concept learning in discrete spaces have been proved by multiple works. However, how cooperative pedagogy can facilitate machine parameter learning hasn't been thoroughly studied. In this paper, we propose a gradient optimization based teacher-aware learner who can incorporate teacher's cooperative intention into the likelihood function and learn provably faster compared with the naive learning algorithms used in previous machine teaching works. We give theoretical proof that the iterative teacher-aware learning (ITAL) process leads to local and global improvements. We then validate our algorithms with extensive experiments on various tasks including regression, classification, and inverse reinforcement learning using synthetic and real data. We also show the advantage of modeling teacher-awareness when agents are learning from human teachers.

## 1 Introduction

Cooperative pedagogy is invoked across language, cognitive development, cultural anthropology, and robotics to explain people's ability to effectively transmit information and accumulate knowledge [19, 64]. As the usage of artificial intelligence and machine learning based systems ratchets up, it is foreseeable that extensive human-computer and agent-agent pedagogical scenarios will occur in the near future [49]. However, there is still a distance away from robots being able to directly teach or learn from humans as efficiently and effectively as humans do. One of the many difficulties is that machine learning and teaching are now usually studied in single-agent frameworks. Most of the prevailing machine learning methods focus on the improvement of **individual learners** and the explanations of how knowledge is obtained focus entirely on each learner's unilateral experiences, either passive observations from a Markov decision process [45, 55], random samples from a data distribution [52, 29], responses of active queries provided by an oracle [3, 53], or demonstrations from an expert [4].

35th Conference on Neural Information Processing Systems (NeurIPS 2021).

Such machine learning framework is diametrically distinctive from human education, in whose context, learning often occurs sequentially instead of in batch, and from intentional messages given by a pedagogical teacher rather than random data from a fixed sampling process [62]. Recently, the advantage of pedagogical teachers over randomly sampled data or optimal task completion trajectories from experts has been shown in machine teaching [10, 70, 71, 40, 18, 44, 11, 12] and in learning from demonstration (LfD) [27, 31]. Nonetheless, compared with human pedagogy, these works still lack a sophisticated student model that can accommodate the teacher's cooperation into his learning and acts differently than learning from passive data. Machine teaching algorithms model a cooperative teacher giving instructions in the format of data examples for continuous parameter [10, 70, 71, 40], Bayesian concept [18, 44] or version space learning [11, 12], but seldom do they consider how learners may interpret differently between the data picked intentionally by the teacher and sampled randomly from the world. Standard LfD takes in demonstrations from an (approximately-) optimal expert to learn the underlying reward function [4]. Hadfield-Menell et al. [27], Ho et al. [31, 32] shows the advantage of using pedagogical rather than optimal demonstrations, yet, in either case, the learners are not aware of the teacher . Shafto et al. [54], Yang et al. [68], Wang et al. [62, 64] move one step further and proposed recursive cooperative inference models having both the teacher and the student reasoning each other, an ability known as theory of mind (ToM) [50, 8]. The first work modeled and predicted human behavior while the latter three managed to integrate ToM into machine learning. Despite the theoretical contribution, their approach [68, 62, 64] is confined to Bayesian concept learning with finite hypothesis space, in which the Sinkhorn scaling [57] is tractable. It is unclear how to apply their algorithms to settings involving continuous hypothesis spaces, such as learning neural networks.

In this paper, we study how to integrate the cooperative essence of pedagogy into machine parameter learning and propose a teacher-aware learner who learns significantly faster than a naive learner, given an iterative machine teacher [40, 41]. The learner estimates the teacher's data selection process with distribution and corrects his likelihood function with this estimation to accommodate the teacher's intention. Maximizing the new likelihood enables the learner to utilize both explicit information from the selected data and implicit information suggested by the pedagogical context. We theoretically proved the improvement brought by the learner's teacher awareness and justified our results with various experiments. We believe that our work can provide insights into both human-machine interactions, such as online education, and machine-machine communications, such as ad-hoc teamwork [9].

Our main contributions are **i)** modeling teacher-awareness for generic gradient optimization based parameter learning; **ii)** theoretically proving the improvement guaranteed by the teacher-aware learner over the naive learner under mild assumptions; **iii)** empirically illustrating the advantage of teacher-awareness learner when interacting with both machine and human teachers.

## 2  Related Work

**Machine Teaching**   Our work is related to machine teaching as we used an iterative machine teacher in our framework. Machine teaching [70, 71, 40, 49] solves the problem of constructing an optimal (usually minimal) dataset according to a target concept such that a student model can learn the target concept based on this dataset. Most of the machine teaching algorithms consider a batch setting, where the teacher designs a minimal dataset and provides it to a learning algorithm in one shot [10, 70, 47, 71, 11]. Iterative machine teaching has also been studied, where the teacher gives out data iteratively and checks the learner's status before selecting the next data [34, 6, 40, 41, 43], but previous works didn't consider teacher-aware learners. There are also works applying machine teaching to inverse reinforcement learning (IRL) and LfD [4, 10, 27, 31, 32, 28]. Our IRL experiments are different from those works as our data are provided iteratively and sequentially. Also, our learner is aware of the teacher's intention. Ho et al. [31, 32] integrated Bayesian rule in LfD to model mutual reasoning between the teacher and the learner, but they mainly used their model to explain human data. A theoretical study of the teaching-complexity of the teacher-aware learners was presented in [73, 17] where the teacher and the learner are aware of their cooperation. Their analysis mainly attends to version-space learners which maintain all hypotheses consistent with the training data, and cannot be applied to hypothesis selection via optimization, such as learning parameters.

Peltola et al. [49] studied machine teaching for an active sequential multi-arm bandit learner. Although they also have a helpful teacher and a teacher-aware learner, their problem setting is different from ours. First, in every round of multi-arm bandit, the learner can actively choose an arm to pull, and

then the teacher provides feedback for that arm, while in our cases, the data batch in each round is sampled randomly and independently from the learner's current status. Second, as the teacher can only give binary (success or not) feedback to the learner, the counterfactual reasoning required for pedagogy is significantly simplified. Besides, they required that the same feature representation for the arms is shared between the teacher and the learner. Also, the learner doesn't have to know the underlying parameter exactly to perform well in multi-arm bandit games, while in most of our cases, the learner tries to match the teacher's model exactly. Fisac et al. [20] used similar formulation to model cooperative value alignment within a human-robot team. They assumed the human knows the robot's value function during interactions, and parameters to be aligned are sparse and low dimensional.

**Learning to Teach** Sharing the same goal as machine teaching, learning to teach (L2T) also seeks a teaching algorithm to improve the learning efficiency of AI. While machine teaching usually models the question as an optimization problem and solves for a closed-form teaching algorithm, works in L2T tend to train the teaching model in the process of the teacher-student interaction with gradient based optimization [67] or reinforcement learning (RL) algorithms [19]. Nevertheless, these works also focus mainly on the teacher algorithm and assume teacher-unaware learners. In some tasks, typically when the student aims to learn a concept in a discrete space, the teacher can track the learner's status by modeling his belief over the concept [54]. As the beliefs within the learner's mind are not usually known by the teacher, the teaching process can be modeled as a partially observable Markov decision process (POMDP) [46], solving the optimal policy of which returns a teaching algorithm [51, 65]. From the teacher's perspective, the unknown part of the environment is the learner's belief, a probability distribution over the concept space, so she has to form another belief over the learner's belief. The intractable modeling of belief over continuous variables usually requires approximation methods such as particle filters [51, 65] to solve, restricting the scope of these algorithms to naive learners and relatively simple learning tasks. Interactive POMDP [23, 66], a probabilistic multi-agent model, can also be used to model cooperative pedagogy with recursive teacher-learner reasoning. However, the nested belief over belief also suffers from intractability and is hard to scale up. If the concept space of the learner is continuous by itself, such as high dimensional continuous parameter spaces in our case, handling the belief over belief becomes difficult.

**Cooperative Bayesian Inference** Shafto et al. [54] studies human pedagogy with examples using Bayesian learning models. The cooperative inference [68, 63, 64] in machine learning also formalizes full recursive reasoning from the perspectives of both the teacher and the learner. Distinctive from their concept learning settings, in this paper, we focus on parameter learning, in which the student has intractable posterior distribution and learns via gradient-based optimization. In addition, [54, 68, 63, 64] only consider the problem of a single interaction between the teacher and learner. Wang et al. [62] proposed a sequential cooperative Bayesian inference algorithm and showed its performance advantage over naive Bayesian learner analytically and empirically. Nevertheless, they were still discussing concept learning with finite and usually small data and hypothesis sets. The Sinkhorn scaling [30, 62, 64] becomes infeasible when dealing with continuous parameter learning.

**Pragmatics Reasoning** Our work is inspired by the study of pragmatics (how context contributes to the meaning) [26, 27] and ToM [50, 8]. The rational speech act (RSA) model raised by Golland et al. [24] and developed in [22, 54, 25, 2] accommodates the idea of using not only the utterance but also the selection of the utterance to understand the speaker. Previous works in these areas are mainly from human action interpretation [21, 60, 36], language emergence [69, 35], linguistics [33, 2, 13] and cognitive science [25, 8] perspectives. To our knowledge, our work is the first to relate pedagogy and recursive reasoning to generic parameter learning and shows a provable improvement. Shafto et al. [54] proposed computational models for more diverse concept spaces, but mainly focus on modeling and predicting human behaviors.

## 3 Background

Finding the optimal way of teaching parameters has been a challenging problem because of the continuous state space and long horizon planning. One common framework is machine teaching [70, 71]. Here, we adopt an iterative variation of machine teaching [40], consisting of three entities: the learner, the teacher and the world. **The world** is defined as a parameter $\omega^*$, fixed and known only by the teacher. Given a model $y = h(x; \omega)$ parameterized by $\omega$, the world is defined as $\omega^* = \arg\min_{\omega \in \Omega} \mathbb{E}_{(x,y) \sim \mathcal{P}(x,y)}[l(h(x; \omega), y)]$, where $\mathcal{P}(x, y)$ is the data distribution in standard

machine learning. Here, $l$ and $h$ can vary across tasks, eg. $l$ can be squared loss for regression, cross-entropy for classification, and negative log-likelihood for IRL [5, 42]. In this paper, we assume $l$ to be a convex function and $h(x; \omega) = h(\langle x, \omega \rangle)$. $h$ can be an identity function for linear regression and softmax function for classification. Thus, we can omit $h$ in the loss function and write $l(\langle x, \omega \rangle, y)$ for short. $l$ and $h$ are common knowledge of the teacher and the learner.

**Representation:** The teacher represents an example as $(x, y)$ while the student represents the same example as $(\tilde{x}, \tilde{y})$ (typically $y = \tilde{y}$ and we use $y$ when there is no ambiguity). The representation $x \in \mathcal{X}$ and $\tilde{x} \in \tilde{\mathcal{X}}$ can be different but deterministically related by an unknown mapping, $\tilde{x} = \mathcal{G}(x)$. Suppose the teacher and the learner use model $h(\langle x, \omega \rangle)$ and $h(\langle \tilde{x}, \nu \rangle)$ respectively, then $\omega^*$ and $\nu^*$ are very likely in different spaces too. This is a common scenario when the teacher and the learner are a human and a robot, or two robots from different factories. As the representation of examples can be complex, such as features extracted by deep neural networks [45, 52, 29], using a linear model $h$ doesn't impinge the expressive power of the overall model. In the rest of the paper, we use $\omega$ for the teacher's parameter and $\nu$ for the learner's if they are from different spaces. Otherwise, we use $\omega$ for both of them. We use $x$ to refer to an example and its teacher representation. We use $\tilde{x}$ for its learner representation. Also, we don't specify the choice of $\mathcal{G}$. Our only assumption about the teacher and the learner's representation will be discussed in Theorem 1.

**Teacher:** In general, the teacher can only communicate with the learner via examples. This restriction doesn't impinge the generality of the machine teaching framework, as the format of the data can be generic, such as demonstration used in the IRL [72, 5, 61, 42]. In this paper, data are provided iteratively. We use $x^t$ to denote the example used in the $t$-th iteration. The teacher aims to provide examples iteratively so that the student parameter $\nu$ converges to its optimum $\nu^*$ as fast as possible. Since the teacher doesn't know $\nu^t$ or $\nu^*$, we let the learner provide some feedback to her in each iteration so that she can track the pedagogy progress (details in Section 4.1).

**Learner:** The learner has an initial parameter $\nu^0$ before learning. At time $t$, he has learning rate $\eta_t$. The learning algorithms for teacher-unaware learners are often simple. For iterative gradient based optimization, the learner usually uses stochastic gradient descent [40, 41, 19, 67]. Suppose the learner receives $(x^t, y^t)$ from the teacher, his iterative update is:

$$\nu^t = \nu^{t-1} - \eta_t \frac{\partial l(\langle \tilde{x}^t, \nu^{t-1} \rangle, y^t)}{\partial \nu^{t-1}}. \tag{1}$$

**Mutual knowledge:** We limit the mutual knowledge between the teacher and the learner, otherwise, the mutual reasoning between the two can theoretically become an infinite recursion. In this paper, we consider a teacher who assumes a naive learner using Eq. (1) to update his model. Meanwhile, the learner knows the teacher selects data deliberately instead of randomly (detailed in the next section). If we define a naive learner as having level-0 recursive reasoning, then the teacher and the teacher-aware learner have level-1 and level-2 recursive reasoning respectively. This level of recursion is very close to human cognitive capability [14, 15] and was also adopted by [49].

To summarize, the loss function $l$, the model $h$, and the naive learner update function are common knowledge to the teacher and the learner. $\omega^*$ and the teaching mechanism are only known by the teacher, while $\nu^t$ and the learning mechanism, i.e. how to update $\nu^t$ given data, are only known by the learner. He knows the teacher intentionally selects helpful data according to her estimation of himself, and the teacher assumes that the learner learns following Eq. (1). For our teacher-aware learner, this assumption is inaccurate, but we'll show how the proposed learner can learn much faster than a naive learner.

## 4 Teacher-Aware Learning

### 4.1 Cooperative Teacher

We first define the teacher whom the learner should be aware of. Let's consider a teacher using the same feature representation as the learner and knowing his parameter in each iteration. [40] termed this kind of teacher as the omniscient teacher, who, in the $t-$th iteration, greedily chooses example

from a data batch $D^t = \{(x_i, y_i) \sim \mathcal{P}(x, y)\}$:

$$(x^t, y^t) = \underset{(x,y) \in D^t}{\arg\min} \left\| \omega^{t-1} - \eta_t \frac{\partial l(\langle x, \omega^{t-1} \rangle, y)}{\partial \omega^{t-1}} - \omega^* \right\|_2^2$$

$$= \underset{(x,y) \in D^t}{\arg\max} \left( -\eta_t^2 \left\| \frac{\partial l(\langle x, \omega^{t-1} \rangle, y)}{\partial \omega^{t-1}} \right\|_2^2 + 2\eta_t \left\langle \omega^{t-1} - \omega^*, \frac{\partial l(\langle x, \omega^{t-1} \rangle, y)}{\partial \omega^{t-1}} \right\rangle \right). \quad (2)$$

The expression after $\arg\max$ in Eq. (2) is defined as the teaching volume $TV_{\omega^*}(x, y|\omega^t)$, which represents the learner's progress in this iteration. It is a trade-off between the difficulty and the usefulness of an example [see 40, sec. 4.1]. Notice that the teacher has no control over $D^t$, which is sampled from the data distribution $\mathcal{P}$ or from a large dataset. She only selects the best example from $D^t$. Given $D^t$ with a mild batch size, e.g. 20, the $\arg\max$ in Eq. (2) can be exactly calculated.

Lessard et al. [38] has proved that, for an omniscient teacher, teaching greedily is sub-optimal. Yet, their findings cannot be directly applied to more practical teaching scenarios. Thus, we keep leveraging the greedy heuristic to model our cooperative teacher and generalize it to a non-omniscient teacher who doesn't fully know the learner in every iteration.

Suppose the teacher neither knows the learner's $\nu^{t-1}$ nor $\nu^*$ and they use different feature representations of the data. To teach cooperatively, she has to imitate the learner's model in her own feature space and use $\omega^*$ to guide the teaching. This can be done approximately if, in every round, the learner gives the inner products of $\nu^{t-1}$ and the data to the teacher as feedback. Given the convexity of the loss function $l$, we have: $\left\langle \omega^{t-1} - \omega^*, \frac{\partial l(\langle x, \omega^{t-1} \rangle, y)}{\omega^{t-1}} \right\rangle \geq l(\langle x, \omega^{t-1} \rangle, y) - l(\langle x, \omega^* \rangle, y)$.
Now, Eq. (2) can be approximated by inner products between the model parameter and the data [40]. Denote the learner's feedback as $\alpha_x = \langle \tilde{x}, \nu^{t-1} \rangle, \forall (x, y) \in D^t, \tilde{x} = \mathcal{G}(x)$, then the teacher will teach as following:

$$\underset{(x,y) \in D^t}{\arg\max} \left( -\eta_t^2 \left\| \frac{\partial l(\alpha_x, y)}{\partial \alpha_x} x \right\|_2^2 + 2\eta_t \big( l(\alpha_x, y) - l(\langle x, \omega^* \rangle, y) \big) \right). \quad (3)$$

It has been shown that cooperative teachers using Eq. (3) can substantially speed up the learning of a standard SGD learner [40]. Nonetheless, only having a cooperative teacher doesn't provide us the most effective interaction between the two agents, as the learner doesn't exploit the fact that the data come from a helpful teacher [54]. In the next section, we introduce a teacher-aware learner.

### 4.2 Teacher-Aware Learner

Now we propose a learner who integrates teacher's pedagogy into his parameter updating process. Suppose we have a distribution $p(x, y|\nu^* = \nu) \propto \exp(-l(\langle \tilde{x}, \nu \rangle, y))$, denoted as $p_\nu(x, y)$. Then, applying gradient descent to $l(\langle \tilde{x}, \nu \rangle, y)$ is equivalent to maximizing $\log p_\nu(x, y)$ wrt. $\nu$. Hence, a learner updating parameters with Eq. (1) can be considered as performing maximum likelihood estimation (MLE) when the data are randomly sampled from $\mathcal{P}(x, y)$.

Nonetheless, in the machine teaching framework, data are no longer randomly sampled from $\mathcal{P}(x, y)$. A teacher-aware learner should rectify his updating rule by considering teacher's helpfulness. Given the dataset $D^t$ at time $t$, the learner can postulate that the teacher is more likely to choose the example she thinks helpful following $p(x, y|\nu^* = \nu, \nu^{t-1}, D^t)$, denoted as $q_\nu(x, y|\nu^{t-1}, D^t)$ for short:

$$q_\nu(x, y|\nu^{t-1}, D^t) = \frac{\exp(\beta_t \widehat{TV}_\nu(\tilde{x}, y|\nu^{t-1}))}{\int_{(x',y') \in D^t} \exp(\beta_t \widehat{TV}_\nu(\tilde{x}', y'|\nu^{t-1}))}, \beta_t \geq 0, \quad (4)$$

$$\text{with } \widehat{TV}_\nu(\tilde{x}, y|\nu^{t-1}) = -\eta_t^2 \left\| \frac{\partial l(\langle \tilde{x}, \nu^{t-1} \rangle, y)}{\partial \nu^{t-1}} \right\|_2^2 + 2\eta_t \big( l(\langle \tilde{x}, \nu^{t-1} \rangle, y) - l(\langle \tilde{x}, \nu \rangle, y) \big). \quad (5)$$

The Boltzmann noisy rationality model [7] indicates that the teacher samples data according to the soft-max of their approximation of the teaching volumes, calculated wrt. her $\nu^*$ and the inner product feedback from the learner. Although, in practice, this estimation is usually different from the teacher's actual example selection distribution, which is a hard-max, corresponding to $\beta_t \to \infty$, maximizing it wrt. $\nu$ can still improve the learning.

The learner now wants to learn a $\nu$, which not only makes $y^t$ more likely to be the correct label of $x^t$, but also $(x^t, y^t)$ more likely to be chosen from $D^t$. Intuitively, given all data in $D^t$ are coherent with the true distribution, the teacher gives $(x^t; y^t)$ but not other examples. With what $\nu$ can the probability of this selection be maximized? So, at every time $t$, the student maximizes $p_\nu(x^t, y^t)$ and $q_\nu(x^t, y^t | \nu^{t-1}, D^t)$ wrt. $\nu$ simultaneously. We can still use gradient descent. Omit $y$ when there is no confusion. Denote $g_x(\gamma) = \frac{\partial l(\langle \tilde{x}, \gamma \rangle, y)}{\partial \gamma}$, then we have (derivation in supplementary Section A.1):

$$\nu^t = \nu^{t-1} - \eta_t \Big( \frac{\partial l(\langle \tilde{x}^t, \nu^{t-1} \rangle, y)}{\partial \nu^{t-1}} - \frac{\partial \log q_\nu(\tilde{x}^t, y^t)}{\partial \nu} \Big|_{\nu = \nu^{t-1}} \Big)$$
$$= \nu^{t-1} - \eta_t g_{x^t}(\nu^{t-1}) - 2\beta_t \eta_t^2 \big( g_{x^t}(\nu^{t-1}) - \mathbb{E}_{x \sim q_{\nu^{t-1}}}[g_x(\nu^{t-1})] \big). \qquad (6)$$

Notice that $\nu^{t-1}$ is a constant in $q_\nu(x^t, y^t | \nu^{t-1}, D^t)$ and the optimization is wrt. $\nu$, which is treated as $\nu^*$ in the calculation. The gradient is computed at $\nu = \nu^{t-1}$. This is equivalent to maximizing a new log-likelihood function $\log \big( p_\nu(x^t, y^t) q_\nu(x^t, y^t | \nu^{t-1}, D^t) \big)$, an approximation of the log probability that $(x^t, y^t)$ being sampled in $D^t$ and then being selected by the teacher given $\nu^* = \nu$. The product is an approximation of $p\big((x^t, y^t), D^t | \nu^{t-1}, \nu^* = \nu\big)$ because the sampling of data in $D^t$ except $(x^t, y^t)$ is regarded as deterministic. When $\beta_t = 0$, i.e. the learner thinks the teacher uniformly samples data, and Eq. (6) becomes regular SGD.

An interpretation of the benefits brought by Eq. (6) is that the learner not only learns from the literal meaning of the example selected by the teacher **(the second term)**, but also compares that example with "also-rans" in $D^t$ **(the third term)**, forming a context incorporating additional information. This is a prevailing phenomenon in human communication, as messages often convey both literal meanings and pragmatic (contributed by the context) meanings [60, 58, 69]. In other words, we can acquire not only explicit information from what others said but also implicit information from what others didn't say. When the message space is finite and known, exact computations of the implicit information becomes tractable. Therefore, in scenarios like human-robot interactions, where robots usually provide predefined user interfaces with a fixed choice of instructions, our algorithm can easily conduct counterfactual reasoning by comparing the user's selected instruction with the others and deliver faster learning than only using the selected one. In Section 5.2, our experiment with humans as the teacher illustrates such an advantage.

One nuance is that if we use $\nu^{t-1}$ as the $\nu$ in Eq. (4), the second term of the teaching volume will be 0. Thus, to better approximate $\nu^*$, in practice, we plug in $\nu^{t-1} - \eta_t \frac{\partial l(\langle \tilde{x}, \nu^{t-1} \rangle, y)}{\partial \nu^{t-1}}$. That is, the learner first updates $\nu^{t-1}$ just like a naive learner. Then he calculates the gradient of $\log q_\nu$ wrt. the new $\nu$ and does an additional gradient descent corresponding to the last term in Eq. (6). Also, in supervised learning settings, the teacher needs to provide labels of the whole dataset for the learner to calculate the expectation. This is a mild requirement easy to be satisfied in practice. In the iterative process, $D^t$ is a mini-batch sampled from a large dataset with a small batch size, say 20 examples. Thus, $\mathbb{E}_{x \sim q_\nu}[g_x(\nu)]$ can be calculated exactly, and, compared with the standard mini-batch gradient descent, the only additional information needed from the teacher is the index of $(x_t, y_t)$. In fact, we can further relax this condition by letting the learner estimate $\mathbb{E}_{x \sim q_\nu}[g_x(\nu)]$ with only a subset $\widehat{D}^t \subseteq D^t$. In our experiments, we show that with only one random unchosen example provided, i.e. $|\widehat{D}^t| = 1$, the teacher-aware learner outperforms the naive learner. See Algorithm 1 for details.

We now prove the teacher-aware learner can always perform better than a naive learner given proper conditions.

**Theorem 1** (Local Improvement). *Denote $\tilde{\nu}^t = \nu^{t-1} - \eta_t g_{x^t}(\nu^{t-1})$. For a specific loss function $l$, given the same learning status $\nu^{t-1}$ and a teacher following Eq. (3), suppose $x^t$ satisfies that $x^t$ itself maximizes $\widehat{TV}_{\tilde{\nu}^t}(x, y | \nu^{t-1})$. Denote $\hat{x}^t$ as the $x \in D^t$ which achieves the second largest $\widehat{TV}_{\tilde{\nu}^t}(x, y | \nu^{t-1})$. Suppose that $\|g_x(\tilde{\nu}^t)\|_2 \leq G$ for any $x \in D^t$. If $\langle \tilde{\nu}^t - \nu^*, g_{x^t}(\tilde{\nu}^t) - g_{\hat{x}^t}(\tilde{\nu}^t) \rangle > 0$, then with large enough $\beta_t$, the teacher-aware learner using Eq. (6) is guaranteed to make no smaller progress than a naive learner using Eq. (1).*

One intuition for the assumption is that the best example selected by the teacher does bring more benefits to the learner than the other examples do. Suppose we have $\hat{\nu}^t = \tilde{\nu}^t - \eta_t g_{\hat{x}^t}(\tilde{\nu}^t)$, then moving from $\hat{\nu}^t$ to $\nu^t$ follows $\eta_t (g_{\hat{x}^t}(\tilde{\nu}^t) - g_{x^t}(\tilde{\nu}^t))$. The assumption $\langle \tilde{\nu}^t - \nu^*, g_{x^t}(\tilde{\nu}^t) - g_{\hat{x}^t}(\tilde{\nu}^t) \rangle = \langle \nu^* - \tilde{\nu}^t, g_{\hat{x}^t}(\tilde{\nu}^t) - g_{x^t}(\tilde{\nu}^t) \rangle > 0$ simply suggests that updating with $x^t$ gives the learner an advantage over updating with $\hat{x}^t$. The advantage points to $\nu^*$ (the two vectors $\nu^* - \tilde{\nu}^t$ and $g_{\hat{x}^t}(\tilde{\nu}^t) - g_{x^t}(\tilde{\nu}^t)$ form an acute angle).

---

**Algorithm 1:** Iterative Teacher-Aware Learning

---

**Input:** Data distribution $\mathcal{D}$, teacher parameter $\omega^*$, learning rate $\eta_t$, teacher estimation scale $\beta_t$

**Result:** $\nu^{(T)}$

1   Randomly initialize student model $\nu^{(0)} \sim \text{Uniform}(N)$; Set $t = 1$ and $T$ as the maximum iteration number;

2   **while** $t < T$ **do**

3      Teacher gets data batch $D^t \sim \mathcal{D}$

4      Learner reports $\alpha_x = \langle \nu^{(t-1)}, \tilde{x} \rangle$ for all $x \in D^t$ to the teacher

5      Teacher selects data for time $t$:

$$(x^t, y^t) = \arg\max_{(x,y) \in D^t} \left( -\eta_t^2 \left\| \frac{\partial l(\alpha_x, y)}{\partial \alpha_x} x \right\|^2 + 2\eta_t \left( l(\alpha_x, y) - l(\langle \omega^*, x \rangle, y) \right) \right)$$

6      Learner uses the selected data $(\tilde{x}^t, y^t)$ and $D^t$ to calculate

7      $\hat{\nu}^{(t)} = \nu^{(t-1)} - \eta_t \frac{\partial l(\langle \tilde{x}^t, \nu^{(t-1)} \rangle, y^t)}{\partial \nu^{(t-1)}}$

8      $\nu^{(t)} = \hat{\nu}^{(t)} - 2\beta_t \eta_t^2 \left( \frac{\partial l(\langle \tilde{x}^t, \hat{\nu}^{(t)} \rangle, y^t)}{\partial \hat{\nu}^{(t)}} - \mathbb{E}_{(\tilde{x}, y) \sim q_{\hat{\nu}^{(t)}}(\tilde{x}, y | \nu^{(t-1)}, D^t)} \left[ \frac{\partial l(\langle \tilde{x}, \hat{\nu}^{(t)} \rangle, y)}{\partial \hat{\nu}^{(t)}} \right] \right)$

9      where $q_{\hat{\nu}^{(t)}}(\tilde{x}, y | \nu^{(t-1)}, D^t) = \dfrac{\exp \left( \beta_t \widehat{TV}_{\hat{\nu}^{(t)}} \left( \tilde{x}, y | \nu^{(t-1)}, D^t \right) \right)}{\sum_{(x', y') \in D^t} \exp \left( \beta_t \widehat{TV}_{\hat{\nu}^{(t)}} \left( \tilde{x}', y' | \nu^{(t-1)}, D^t \right) \right)}$

10      with $\widehat{TV}_{\hat{\nu}^{(t)}}(\tilde{x}, y | \nu^{(t-1)}, D^t)$ defined in Eq. (5).

11      $t = t + 1$

12 **end**

---

**Corollary 2** (Global Improvement). *Under the same condition of Theorem 1, suppose that $\|\partial l(\langle \tilde{x}, \nu \rangle, y)/\partial \nu\|_2^2$ and $l(\langle \tilde{x}, \nu \rangle, y)$ are L-Lipschitz for $x$ with any $\nu$. Suppose the sample set $D^t$ satisfies that for any $x \in D^t$, there exists $x' \in D^t$ such that $\|x' - x\|_2 \leq \epsilon/(TL(\eta_t^2 + 4\eta_t))$ for any $t$, where $T$ is the total number of iterations. Then if the inequality*

$$\|\nu_1 - \nu^*\|_2^2 - \max_{(x,y) \in D^t} \widehat{TV}_{\nu^*}(x, y | \nu_1) \leq \|\nu_2 - \nu^*\|_2^2 - \max_{(x,y) \in D^t} \widehat{TV}_{\nu^*}(x, y | \nu_2) \tag{7}$$

*holds for any $\nu_1, \nu_2$ that satisfy $\|\nu_1 - \nu^*\|_2^2 \leq \|\nu_2 - \nu^*\|_2^2$, then with the same parameter initialization, learning rate and a teacher following Eq. (3), a teacher-aware learner can always converge not slower than a naive learner up to $\epsilon$ error.*

To guarantee that $\|x' - x\|_2 \leq \epsilon/(TL(\eta_t^2 + 4\eta_t))$ for any $x \in D$, we need the subset $D^t \subseteq D$ to be 'uniform distributed' on $D$. To achieve this goal, we can uniformly sample point $x \in D$ and let $D^t$ to be the set of these points. It is easy to verify that the 'uniform distributed' property holds with high probability when $|D^t|$ is large enough. Meanwhile, Eq. (7) in Corollary 2 is defined as teaching monotonicity in Liu et al. [40], and they proved that the squared loss satisfies teaching monotonicity given a dataset $D = \{x \in \mathbb{R}^d, \|x\| \leq R\}$ [see 40, proposition 3]. The main difference between Eq. (7) and that in Liu et al. [40] is that Eq. (7) works for the non-omniscient teacher setting, while Liu et al. [40] focuses on the omniscient teacher setting. Detailed proofs of the theories can be found in Section A of our supplementary.

## 5 Experiments

### 5.1 Machine Teacher

To justify the effectiveness of ITAL, we compared it with iterative machine teaching (IMT) with a naive learner on regression, classification, and IRL tasks. The coverage of squared loss, cross-entropy loss, and negative log-likelihood proves the robustness of our algorithm on various selections of $l$. For regression tasks, we measured the performance using the difference between $\|\omega^t - \omega^*\|_2$ and the mean squared loss of the test set. For the classification task, we measured the difference, the cross-entropy loss, and the classification accuracy of the test set. For online IRL problems, we measured the parameter difference, the total variance between the teacher's and the learner's policies, and the average rewards achieved by the learner. The feature dimension of the teachers can be

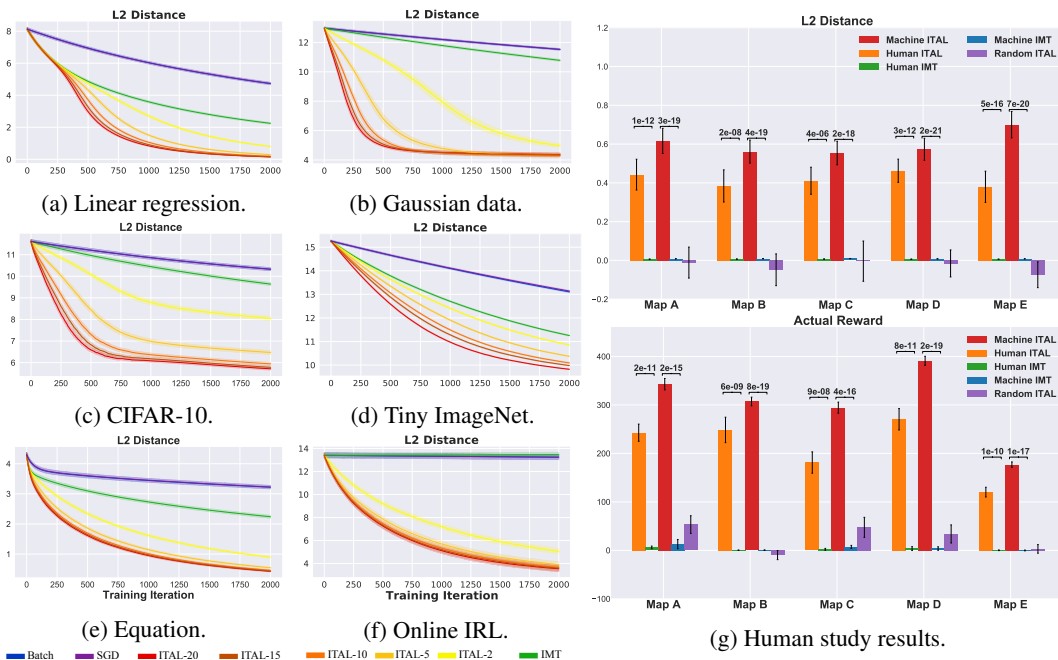

(a) Linear regression.

(b) Gaussian data.

(c) CIFAR-10.

(d) Tiny ImageNet.

(e) Equation.

(f) Online IRL.

(g) Human study results.

Batch    SGD    ITAL-20    ITAL-15    ITAL-10    ITAL-5    ITAL-2    IMT

Figure 1: **Fig. 1a-Fig. 1f**: Cooperative teacher results. Our method always gives a substantial improvement over IMT, showing the effect of teacher-awareness. Within 2000 steps, ITAL already show convergence, while a naive learner only learns to a limited extent in most tasks. **Fig. 1g**: In the top plot, the height of each bar represents the **decrease** of the L2-distance between the learner's reward parameter and the ground-truth parameter. In the bottom, the height represents the accumulated reward. Paired t-tests were conducted between Human (Machine) ITAL & IMT respectively.

different from that of the learners in some experiments. In Fig. 1, we show the results of the teacher having a smaller feature dimension than the learner does. We show the opposite in the supplementary Section B. **Batch** means the learner uses all the data in the mini-batch to calculate the mean gradient. **SGD** means the learner randomly selects an example in the mini-batch to calculate the gradient. **ITAL-$M$s** represent our algorithm with $M$ indicates $|\widehat{D}^t|$. The mini-batch $D^t$ is randomly sampled at every step with batch size 20. The learning rate is `1e-3` for all the experiments. $\beta_t$ is in the scale of `1e4`, varying for different settings. We grid search $\beta_t$ starting from `1e4` and use the largest one inducing Eq. (4) that is no longer a delta function. We ran each experiment with 20 different random seeds to calculate the mean and its standard error, shown in Fig. 1.

It can be seen that using the full mini-batch gives almost identical learning performance as using only one random sample from it. IMT has noticeable but limited improvements comparing with Batch and SGD, suggesting not necessarily substantial advantage brought by the helpful teacher. ITAL, on the other hand, significantly outperforms all other baselines, even with only 2 data points as the approximation of the full mini-batch. The learner modeled by these baselines only learns from the examples, but when the examples are no longer acquired randomly but from an intentional teacher, the example selection of the teacher also conveys a large amount of information. In particular, the teacher-aware learner can absorb information from not only the selected examples but also the unselected ones. As the learner has access to more unselected examples, he has a better approximation of the teaching process and learns more efficiently. Additional experimental details can be found in supplementary Section B.

### 5.1.1 Supervised Learning

**Linear Models on Synthetic Data:** In these experiments, we explored the convergence of our method in linear regression and multinomial logistic regression. For linear regression, we randomly generated a $M$-dimensional vector and a bias term as the $\omega^*$, and $X \in \mathbb{R}^{N \times (d+1)}$ as the training set, with the last column being all 1s. The labels are $Y = X\omega^*$. For the classification task, we randomly generated $K$ points in the $d$-dimensional space, each of which is used as the mean of a normal distribution. Then we sampled $N/K$ points from each Gaussian distribution together as the

training data. The labels are the indices of these distributions. With these data, we trained a logistic regression model using Scikit-learn [48], and used the coefficients as the teacher's $\omega^*$. We used a random orthogonal projection matrix to generate the teacher's feature space from student's. At every step, a subset of the training data is randomly selected as the mini-batch. The data points in that mini-batch along with their labels and the index of the data selected by the teacher are sent to the student. Details of the data generation can be found in supplementary Section B.1.

**Linear Classifiers on Natural Image Datasets:** We further evaluated our teacher-aware learner on image datasets, CIFAR-10 [37] and Tiny ImageNet [1] (an adaptation of ImageNet [16] used in Stanford 231n with 200 classes and 500 images in each class). In these experiments, the teacher tried to teach the parameters of the last fully connected (FC) layer in a convolutional neural network (CNN) trained on the dataset. We trained 3 baseline CNNs independently to do CIFAR-10 10-class and ImageNet 200-class classification. All of them achieved reasonable accuracy ($\geq 82\%$ for CIFAR-10 and $\geq 58\%$ top-1, $\geq 85\%$ top-5 for ImageNet). For CIFAR-10, we trained three different types of CNNs, CNN-6/9/12. For ImageNet we used VGG-13/16/19 [56]. The features fed into the last FC layers are extracted to be the teaching dataset. The learner's feature is from CNN-9/VGG-16 and we set the teacher as either CNN-6/12 or VGG-13/19. Details about CIFAR-10 and Tiny ImageNet experiments are in B.3 and Section B.4 respectively.

**Linear Regression for Equation Simplification:** In this experiment, we learned a linear value function that can be used to guide action selections. Given polynomial equations with fraction coefficients and unmerged terms, we want to simplify them into cleaner forms with all the terms merged correctly, all the coefficients rescaled to integers without common factors larger than 1, and all the terms sorted by the descending power. For example, equation $-\frac{1}{2}x^2y + \frac{1}{3}xy = -\frac{1}{5}y^3 + \frac{1}{3}x^2y$ will be simplified to $-25x^2y + 10xy + 6y^3 = 0$. We defined a set of equation editing actions and a set of simplification rules. For a given equation, we applied the rules, recorded every editing action, and collected a simplifying trajectory. With all the trajectories of the training equations, we trained a value function by assuming that the value monotonically increases in each trajectory. Then the teacher tried to teach the student this value function. We used three different feature dimensions: 40D, 45D, and 50D. The learner always used 45D, and the teacher used 40D or 50D. Details can be found in Section B.5.

### 5.1.2 Online Inverse Reinforcement Learning

In this experiment, we changed from labeled data in standard supervised learning to demonstrations in IRL. The learner wanted to learn the parameter for a linear reward function $r(s, \omega^*)$ so that the likelihood of the demonstrations is maximized [5, 61, 42]. One challenge is that the $\max$ function in Bellman equations [59] is non-differentiable. Thus, we approximated $\max$ with soft-max, namely: $\max(a_0, ..., a_n) \approx \frac{\log(\sum_{i=0}^{n} \exp ka_i)}{n}$, with $k$ controlling the level of approximation and leveraged the online Bellman gradient iteration [39]. The IRL environment is an $8 \times 8$ map, with a randomly generated reward assigned to each grid. If we encode each grid using a one-hot vector, then the reward parameter is a 64D vector with the $i$-th entry corresponding to the reward of the $i$-th grid. The agent can go up, down, left, or right in each grid. All demonstrations are in the format of $(s, a)$, where $s$ indicates a grid and $a$ an action demonstrated in that grid. The teacher uses a shuffled map encoding. For instance, if the first grid is $[1, 0, ..., 0]$ to the learner, then it became $[0, ..., 0, 1, 0, ...]$ to the teacher. Details are included in Section B.6.

### 5.1.3 Adversarial Teacher

In addition to the cooperative setting that we assumed throughout the discussion above, we also explored if the learner can still learn given an adversarial teacher. An adversarial teacher doesn't mean that she gives fake data to the student, but she uses $\arg\min$ in Eq. (4) instead of using $\arg\max$. That is, she always chooses the least helpful data for the learner. Hence, a learner, being aware of this unhelpful pedagogy, will adjust Eq. (3) accordingly by using $\beta_t \leq 0$. We redid all previous experiments with an adversarial teacher and showed that our learner can still learn effectively given an adversarial teacher, while a naive learner barely improves (see Fig. 14 in Section B.7). This experiment justifies the universal utility of modeling the teacher's intention regardless of the informativeness of the teaching examples.

## 5.2 Human Teacher

In the previous section, we showed that teacher-awareness substantially accelerates learning, given a machine teacher. In this section, we further investigate if our teacher-aware learner can show an advantage in scenarios where humans play the role of the cooperative teacher. We hypothesize that despite the discrepancy between the pedagogical pattern of human and machine teachers, our learner can still benefit from his teacher-awareness modeled with Eq. (4).

We conducted a proof-of-concept human study with a similar but simplified version of the IRL experiments in Section 5.1.2. To better suit human participants, we first change the maps from $8 \times 8$ to $5 \times 5$. Second, instead of assigning random continuous rewards to the grids, we color them with white, blue, and red, representing neutral, bad, and good tiles. In each teaching session, a participant is given one of the five reward maps as the ground truth and a randomly initialized learner to be taught. Then, the participant will be asked to teach the learner about the ground truth reward in each grid by providing $(s, a)$ examples as in Section 5.1.2. In each time step, we construct the examples by randomly sampling a set of 10 grids and drawing an arrow on each sampled grid indicating which direction the learner should go to if in that grid. The human teacher is asked to choose the most helpful arrow given the learner's current reward map. The map configurations are in Fig. 15 in Section B.8. A similar map setup for reward teaching was used by Ho et al. [31]. Every participant will teach both the naive and the teacher-aware learner about the same map. We then run a paired sample t-test to compare the learning effect of the two types of learners. We show the improvement of the L2-distance between the learner's reward parameters and the ground-truth reward parameters and the accumulated reward in Fig. 1g. Comparison of policy total variance and learning curves are included in Section B.8 of the supplementary.

For all maps, the ITAL method has a significant ($p$-value $\approx 0$) advantage over its IMT counterparts. The human ITALs all perform worse than machine ITALs. This is as expected as we directly reuse the machine teacher model to simulate humans. There is no guarantee that all the participants follow the same teaching pattern as the machine teacher, or even have a consistent teaching pattern at all. Yet, we still manage to grasp human cooperation to some extent. To illustrate the influence of the teacher model, we also teach the ITAL learner with a random teacher, who samples the example uniformly every time and is not cooperative at all. As shown in Fig. 1g, this combination doesn't benefit the learner, because the mismatch between the imagined cooperative teacher and the actual random teacher will very likely introduce over-interpretation of the examples. To summary, these results justify that human teachers do have cooperative (contrary to uniform) pedagogy patterns and the current teacher-aware model can take advantage of them. Finding a comprehensive and accurate human-robot communication model will be an open question for future works.

## 6 Discussion and Conclusions

Pedagogy has a profound cognitive science background, but it hasn't received much attention in machine learning works until recently. In this paper, we integrate pedagogy with parameter learning and propose a teacher-aware learning algorithm. Our algorithm changes the model update step for the gradient learner to accommodate the intention of the teacher. We provide theoretical and empirical evidence to justify the advantage of the teacher-aware learner over the naive learner.

To be aware of the teacher, the learner needs an accurate estimation of the teaching model. In many cases, such a model is not directly accessible, e.g. when there is a human-in-the-loop. In this paper, we model the teacher in a heuristic manner. Our human study proved the generality of this model, especially when the learner only assumes a sub-optimal teacher with Boltzmann rationality. In future work, a more advanced teacher model should be investigated, acquired through task-specific data and/or interactions between the agents. Another limitation of our work is that, in our current setting, the learner's feedback is restricted to be inner products. A more generic message space can be leveraged to develop comprehensive learning as a bidirectional communication platform. We believe our work illustrates the promising benefits of accommodating human pedagogy into machine learning algorithms and approaching learning as a multi-agent problem.

## Acknowledgments and Disclosure of Funding

The work was supported by DARPA XAI project N66001-17-2-4029, ONR MURI project N00014-16-1-2007 and NSF DMS-2015577. We would like to thank Yixin Zhu, Arjun Akula from the UCLA Department of Statistics and Prof. Hongjing Lu from the UCLA Psychology Department for their help with human study design, and three anonymous reviewers for their constructive comments.

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
