# Iterative Teacher-Aware Learning Supplementary Material

**Luyao Yuan**[1]
yuanluyao@ucla.edu

**Dongruo Zhou**[1]
drzhou@cs.ucla.edu

**Juhong Shen**[2]
jhshen@ucla.edu

**Jingdong Gao**[1]
mxuan@ucla.edu

**Jeffrey L. Chen**[1]
jlchen0@ucla.edu

**Quanquan Gu**[1]
qgu@cs.ucla.edu

**Ying Nian Wu**[3]
ywu@stat.ucla.edu

**Song-Chun Zhu**[1,3]
sczhu@stat.ucla.edu

[1]Department of Computer Science, [2]Department of Mathematics, [3]Department of Statistics
University of California, Los Angeles
[4]Beijing Institute for General Artificial Intelligence (BIGAI)

## Contents

## A  Proofs and Derivations

### A.1  Gradient Derivation

$$\widehat{TV}_\nu(\tilde{x}, y | \nu^{t-1}) = -\eta_t^2 \left\| \frac{\partial l(\langle \tilde{x}, \nu^{t-1} \rangle, y)}{\partial \nu^{t-1}} \right\|_2^2 + 2\eta_t \Big( l\big(\langle \tilde{x}, \nu^{t-1} \rangle, y\big) - l\big(\langle \tilde{x}, \nu \rangle, y\big) \Big)$$

$$\text{Denote } g_x(\nu) = \frac{\partial l(\langle \tilde{x}, \nu \rangle, y)}{\partial \nu}$$

$$\frac{\partial \log q_\nu(\tilde{x}^t, y^t | \nu^{t-1}, D)}{\partial \nu} = \frac{\partial}{\partial \nu} \left( \beta_t \widehat{TV}_\nu(x^t, y^t | \nu^{t-1}) - \log \int_{(x,y) \in D} \exp\left( \beta_t \widehat{TV}_\nu(x, y | \nu^{t-1}) \right) \right)$$

$$= -2\beta_t \eta_t g_{x^t}(\nu) + \frac{2\beta_t \eta_t \int_{(x,y) \sim D} g_x(\nu) \exp(-\beta_t \widehat{TV}_\nu(x, y | \nu^{t-1}))}{\int_{(x,y) \in D} \exp\left( -\beta_t \widehat{TV}_\nu(x, y | \nu^{t-1}) \right)}$$

$$= -2\beta_t \eta_t (g_{x^t}(\nu) - \mathbb{E}_{x \sim q_\nu}[g_x(\nu)])$$

## A.2   Proof of Theorem 1

For simplicity, let $g_x$ denote $g_x(\tilde{\nu}^t)$ in this proof. First we provide an intuition for the assumption. Suppose we have $\hat{\nu}^t = \tilde{\nu}^t - \eta_t g_{\hat{x}^t}$, then moving from $\hat{\nu}^t$ to $\nu^t$ follows $\eta_t(g_{\hat{x}^t} - g_{x^t})$. The assumption $\langle \tilde{\nu}^t - \nu^*, g_{x^t} - g_{\hat{x}^t} \rangle = \langle \nu^* - \tilde{\nu}^t, g_{\hat{x}^t} - g_{x^t} \rangle > 0$ simply suggests that updating with $x^t$ gives the learner an advantage over updating with $\hat{x}^t$. The advantage points to $\nu^*$ (the two vectors $\nu^* - \tilde{\nu}^t$ and $g_{\hat{x}^t} - g_{x^t}$ form an acute angle).

Next, we start the proof. We need the following lemma:

**Lemma 1.** *Denote $\hat{x}^t$ as the $x$ which achieves the second largest $\widehat{TV}_{\nu^*}(\tilde{x}, y | \nu^{t-1})$. Suppose that $\langle \tilde{\nu}^t - \nu^*, g_{x^t} - g_{\hat{x}^t} \rangle > 0$, then there exists $\alpha > 0$ such that with large enough $\beta_t$, we have*

$$\left\| \beta_t \eta_t^2 (g_{x^t} - \mathbb{E}_{x \sim q_{\tilde{\nu}^t}}[g_x]) \right\|_2 \leq \alpha \| \tilde{\nu}^t - \nu^* \|_2 \tag{S-1}$$

*and*

$$\langle \tilde{\nu}^t - \nu^*, g_{x^t} - \mathbb{E}_{x \sim q_{\tilde{\nu}^t}}[g_x] \rangle \geq \alpha \| \tilde{\nu}^t - \nu^* \|_2 \left\| g_{x^t} - \mathbb{E}_{x \sim q_{\tilde{\nu}^t}}[g_x] \right\|_2. \tag{S-2}$$

*Proof of Lemma 1.* We set $\alpha = \langle \tilde{\nu}^t - \nu^*, g_{x^t} - g_{\hat{x}^t} \rangle / (2 \| \tilde{\nu}^t - \nu^* \|_2 \| g_{x^t} - g_{\hat{x}^t} \|_2) > 0$.

First we show that $\left\| \beta_t \eta_t^2 (g_{x^t} - \mathbb{E}_{x \sim q_{\tilde{\nu}^t}}[g_x]) \right\|_2 \leq \alpha \| \tilde{\nu}^t - \nu^* \|_2$. For simplicity we denote $s(x) = \widehat{TV}_{\tilde{\nu}^t}(\tilde{x}, y | \nu^{t-1})$. Then by assumption on the selection of $x^t$ we have $x^t = \arg\max_{x \in D^t} s(x)$ and

$$(g_{x^t} - \mathbb{E}_{x \sim q_{\tilde{\nu}^t}}[g_x]) \int_{x' \in D} \exp(\beta_t s(x'))$$

$$= \left( g_{x^t} - \frac{\int_{x' \in D} \exp(\beta_t s(x')) g_{x'}}{\int_{x' \in D} \exp(\beta_t s(x'))} \right) \int_{x' \in D} \exp(\beta_t s(x'))$$

$$= \exp(\beta_t s(x^t))[g_{x^t} - g_{x^t}] + \exp(\beta_t s(\hat{x}^t))[g_{x^t} - g_{\hat{x}^t}] + \sum_{x \neq x^t, \hat{x}^t} \exp(\beta_t s(x))[g_{x^t} - g_x]$$

$$= \exp(\beta_t s(\hat{x}^t))[g_{x^t} - g_{\hat{x}^t}] + \sum_{x \neq x^t, \hat{x}^t} \exp(\beta_t s(x))[g_{x^t} - g_x]. \tag{S-3}$$

Therefore, denote $\xi_{t-1} = s(\hat{x}^t) - s(x^t) < 0$, we have that when $\beta_t \to \infty$,

$$\beta_t \left\| g_{x^t} - \mathbb{E}_{x \sim q_{\tilde{\nu}^t}}[g_x] \right\|_2$$

$$\leq \beta_t \left\| g_{x^t} - \mathbb{E}_{x \sim q_{\tilde{\nu}^t}}[g_x] \right\|_2 \int_{x' \in D} \exp(\beta_t[s(x') - s(x^t)])$$

$$= \beta_t \exp(-\beta_t s(x^t)) \left\| (g_{x^t} - \mathbb{E}_{x \sim q_{\tilde{\nu}^t}}[g_x]) \int_{x' \in D} \exp(\beta_t s(x')) \right\|_2$$

$$= \beta_t \left\| \exp(\beta_t[s(\hat{x}^t) - s(x^t)])[g_{x^t} - g_{\hat{x}^t}] + \sum_{x \neq x^t, \hat{x}^t} \exp(\beta_t[s(x) - s(x^t)])[g_{x^t} - g_x] \right\|_2$$

$$\leq \beta_t \sum_{x \neq x^t} \left\| \exp(\beta_t[s(x) - s(x^t)])[g_{x^t} - g_x] \right\|_2$$

$$\leq \beta_t |D| \exp\left( \beta_t \xi_{t-1} \right) \max_{x' \in D} \| g_{x^t} - g_{x'} \|_2$$

$$\leq \beta_t |D| \exp\left(\beta_t \xi_{t-1}\right) \max_{x' \in D} \left(\|g_{x^t}\|_2 + \|g_{x'}\|_2\right)$$

$$= 2\beta_t |D| \exp\left(\beta_t \xi_{t-1}\right) G$$

$$\to 0,$$

where the first inequality holds due to the fact $s(x') < s(x^t)$ for any $x' \in D$, the second inequality holds due to triangle inequality, the third inequality holds due to the facts $\left(s(x) - s(x^t)\right) < \xi_{t-1}$ for any $x \neq x^t$, the fourth inequality holds due to the assumption that $\|g_x\|_2 \leq G$, the last line holds due to the fact that $x \exp(ax) \to 0$ for $a < 0$ and $x \to \infty$. Therefore, taking large enough $\beta_t$, we have

$$\left\| \beta_t \eta_t^2 (g_{x^t} - \mathbb{E}_{x \sim q_{\tilde{\nu}^t}}[g_x]) \right\|_2 \leq \alpha \|\tilde{\nu}^t - \nu^*\|_2.$$

Next we show that $\langle \tilde{\nu}^t - \nu^*, g_{x^t} - \mathbb{E}_{x \sim q_{\tilde{\nu}^t}}[g_x] \rangle \geq \alpha \|\tilde{\nu}^t - \nu^*\|_2 \|g_{x^t} - \mathbb{E}_{x \sim q_{\tilde{\nu}^t}}[g_x]\|_2$. From (S-3) we have

$$(g_{x^t} - \mathbb{E}_{x \sim q_{\tilde{\nu}^t}}[g_x]) \exp(-\beta_t s(\hat{x}^t)) \int_{x' \in D} \exp(\beta_t s(x'))$$

$$= g_{x^t} - g_{\hat{x}^t} + \underbrace{\sum_{x \neq x^t, \hat{x}^t} \exp(\beta_t [s(x) - s(\hat{x}^t)])[g_{x^t} - g_x]}_{g(\beta_t)},$$

For $g(\beta_t)$, denote $\hat{\xi}_{t-1} = \min_{x \neq x^t, \hat{x}^t} [s(x) - s(\hat{x}^t)] < 0$. Then when $\beta_t \to \infty$, we have

$$\|g(\beta_t)\|_2 \leq \sum_{x \neq x^t, \hat{x}^t} \left\| \exp(\beta_t [s(x) - s(\hat{x}^t)])[g_{x^t} - g_x] \right\|_2$$

$$\leq |D| \exp(\beta_t \hat{\xi}_{t-1}) \max_{x \in D} \|g_{x^t} - g_x\|_2$$

$$\leq 2G|D| \exp(\beta_t \hat{\xi}_{t-1})$$

$$\to 0, \tag{S-4}$$

where the first inequality holds due to triangle inequality, the second inequality holds due to the fact $s(x) - s(\hat{x}^t) < \hat{\xi}_{t-1}$, the third inequality holds due to the assumption that $\|g_x\|_2 \leq G$, the last line holds because $\exp(-x) \to 0$ when $x \to \infty$. Thus when $\beta_t \to \infty$, we have

$$\frac{g_{x^t} - \mathbb{E}_{x \sim q_{\tilde{\nu}^t}}[g_x]}{\|g_{x^t} - \mathbb{E}_{x \sim q_{\tilde{\nu}^t}}[g_x]\|_2} = \frac{(g_{x^t} - \mathbb{E}_{x \sim q_{\tilde{\nu}^t}}[g_x]) \exp(-\beta_t s(\hat{x}^t)) \int_{x' \in D} \exp(\beta_t s(x'))}{\left\| (g_{x^t} - \mathbb{E}_{x \sim q_{\tilde{\nu}^t}}[g_x]) \exp(-\beta_t s(\hat{x}^t)) \int_{x' \in D} \exp(\beta_t s(x')) \right\|_2}$$

$$= \frac{g_{x^t} - g_{\hat{x}^t} + g(\beta_t)}{\|g_{x^t} - g_{\hat{x}^t} + g(\beta_t)\|_2}$$

$$\to \frac{g_{x^t} - g_{\hat{x}^t}}{\|g_{x^t} - g_{\hat{x}^t}\|_2},$$

where the last line holds due to $g(\beta_t) \to 0$ from (S-4). Therefore, we know that for large enough $\beta_t$, we have

$$\left| \left\langle \frac{\tilde{\nu}^t - \nu^*}{\|\tilde{\nu}^t - \nu^*\|_2}, \frac{g_{x^t} - \mathbb{E}_{x \sim q_{\tilde{\nu}^t}}[g_x]}{\|g_{x^t} - \mathbb{E}_{x \sim q_{\tilde{\nu}^t}}[g_x]\|_2} \right\rangle - \left\langle \frac{\tilde{\nu}^t - \nu^*}{\|\tilde{\nu}^t - \nu^*\|_2}, \frac{g_{x^t} - g_{\hat{x}^t}}{\|g_{x^t} - g_{\hat{x}^t}\|_2} \right\rangle \right| \leq \alpha,$$

Finally, due to the fact $\langle \tilde{\nu}^t - \nu^*, g_{x^t} - g_{\hat{x}^t} \rangle = 2\alpha \|\tilde{\nu}^t - \nu^*\|_2 \|g_{x^t} - g_{\hat{x}^t}\|_2$ from (S-2), we have

$$\left\langle \frac{\tilde{\nu}^t - \nu^*}{\|\tilde{\nu}^t - \nu^*\|_2}, \frac{g_{x^t} - \mathbb{E}_{x \sim q_{\tilde{\nu}^t}}[g_x]}{\|g_{x^t} - \mathbb{E}_{x \sim q_{\tilde{\nu}^t}}[g_x]\|_2} \right\rangle \geq \left\langle \frac{\tilde{\nu}^t - \nu^*}{\|\tilde{\nu}^t - \nu^*\|_2}, \frac{g_{x^t} - g_{\hat{x}^t}}{\|g_{x^t} - g_{\hat{x}^t}\|_2} \right\rangle - \alpha = \alpha.$$

$\square$

Now we prove the main theorem.

*Proof of Theorem 1.* The naive learner and the teacher-aware learner, after receiving $(x^t, y^t)$, will update their model to $\tilde{\nu}^t = \left(\nu^{t-1} - \eta_t g_{x^t}\right)$ and $\nu^t = \left(\nu^{t-1} - \eta_t g_{x^t} - 2\beta_t \eta_t^2 (g_{x^t} - \mathbb{E}_{x \sim q_{\tilde{\nu}^t}}[g_x])\right)$ respectively. Then with large enough $\beta_t$, we have

$$\|\nu^t - \nu^*\|_2^2$$

$$= \left\| \tilde{\nu}^t - \nu^* - 2\beta_t \eta_t^2 (g_{x^t} - \mathbb{E}_{x \sim q_{\tilde{\nu}^t}}[g_x]) \right\|_2^2$$

$$= \|\tilde{\nu}^t - \nu^*\|_2^2 - 4\langle \tilde{\nu}^t - \nu^*, \beta_t \eta_t^2 (g_{x^t} - \mathbb{E}_{x \sim q_{\tilde{\nu}^t}}[g_x]) \rangle + 4 \left\| \beta_t \eta_t^2 (g_{x^t} - \mathbb{E}_{x \sim q_{\tilde{\nu}^t}}[g_x]) \right\|_2^2$$

$$\leq \|\tilde{\nu}^t - \nu^*\|_2^2 - 4\alpha \|\tilde{\nu}^t - \nu^*\|_2 \left\| \beta_t \eta_t^2 (g_{x^t} - \mathbb{E}_{x \sim q_{\tilde{\nu}^t}}[g_x]) \right\|_2 + 4 \left\| \beta_t \eta_t^2 (g_{x^t} - \mathbb{E}_{x \sim q_{\tilde{\nu}^t}}[g_x]) \right\|_2^2$$

$$\leq \|\tilde{\nu}^t - \nu^*\|_2^2 - 4\alpha \|\tilde{\nu}^t - \nu^*\|_2 \left\| \beta_t \eta_t^2 (g_{x^t} - \mathbb{E}_{x \sim q_{\tilde{\nu}^t}}[g_x]) \right\|_2$$
$$\qquad + 4\alpha \|\tilde{\nu}^t - \nu^*\|_2 \left\| \beta_t \eta_t^2 (g_{x^t} - \mathbb{E}_{x \sim q_{\tilde{\nu}^t}}[g_x]) \right\|_2$$

$$= \|\tilde{\nu}^t - \nu^*\|_2^2,$$

where the first inequality holds due to (S-2) in Lemma 1, the second inequality holds due to (S-1) in Lemma 1. $\qquad \square$

### A.3 Proof of Corollary 2

*Proof.* Let $\nu_a$ and $\nu_b$ be the model parameter of the naive learner and the teacher-aware learner. Denote $\widehat{TV}_{\nu^*}^E(\nu) = \max_{x \in E} \widehat{TV}_{\nu^*}(x, y | \nu)$, $E$ is some dataset. Let $x^*$ denote the argmax of $\max_{x \in D} \widehat{TV}_{\nu^*}(x, y | \nu)$, then by the assumption on $D^t$, we know that there exists $x'$ such that $\|x' - x^*\|_2 \leq \epsilon / (TL(\eta_t^2 + 4\eta_t))$. Then we have

$$\widehat{TV}_{\nu^*}^D(\nu) - \widehat{TV}_{\nu^*}^{D^t}(\nu)$$
$$= \max_{x \in D}(-\eta_t^2 g_x(\nu) + 2\eta_t(l(\nu, x) - l(\nu^*, x)) - \max_{x \in D^t}(-\eta_t^2 g_x(\nu) + 2\eta_t(l(\nu, x) - l(\nu^*, x))$$
$$\leq (-\eta_t^2 g_{x^*}(\nu) + 2\eta_t(l(\nu, x^*) - l(\nu^*, x^*)) - (-\eta_t^2 g_{x'}(\nu) + 2\eta_t(l(\nu, x') - l(\nu^*, x'))$$
$$\leq L(\eta_t^2 \|x^* - x'\|_2 + 4\eta_t \|x^* - x'\|_2) \qquad\qquad (L\text{-Lipschitz})$$
$$\leq \epsilon/T. \qquad\qquad\qquad\qquad\qquad\qquad\qquad\qquad\qquad\qquad\qquad (S\text{-}5)$$

Now we prove that $\|\nu_b^t - \nu^*\|_2^2 \leq \|\nu_a^t - \nu^*\|_2^2 + t/T\epsilon$ for all $1 \leq t \leq T$. As $\nu^0$ is the same for both learners, knowing theorem 1, we have $\|\nu_b^1 - \nu^*\|_2^2 \leq \|\nu_a^1 - \nu^*\|_2^2 + 1/T\epsilon$. Suppose $\|\nu_b^t - \nu^*\|_2^2 \leq \|\nu_a^t - \nu^*\|_2^2 + t/T\epsilon$, then we have

$$\|\nu_b^{t+1} - \nu^*\|$$
$$\leq \|\nu_b^t - \nu^*\|_2^2 - TV_{\nu^*}^{D^t}(\nu_b^t) \qquad\qquad\qquad\qquad \text{(Theorem 1)}$$
$$\leq \|\nu_b^t - \nu^*\|_2^2 - \widehat{TV}_{\nu^*}^{D^t}(\nu_b^t) \qquad\qquad\qquad\qquad \text{(convexity of } l)$$
$$\leq \|\nu_b^t - \nu^*\|_2^2 - \widehat{TV}_{\nu^*}^{D}(\nu_b^t) + \epsilon/T \qquad\qquad \widehat{TV}_{\nu^*}^D(\nu) - \widehat{TV}_{\nu^*}^{D^t}(\nu) \leq \epsilon$$
$$\leq \|\nu_a^t - \nu^*\|_2^2 - \widehat{TV}_{\nu^*}^{D}(\nu_a^t) + (t+1)/T\epsilon \qquad\qquad \text{(condition)}$$
$$\leq \|\nu_a^t - \nu^*\|_2^2 - \widehat{TV}_{\nu^*}^{D^t}(\nu_a^t) + (t+1)/T\epsilon \qquad\qquad \widehat{TV}_{\nu^*}^D(\nu) - \widehat{TV}_{\nu^*}^{D^t}(\nu) \geq 0$$
$$= \|\nu_a^{t+1} - \nu^*\|_2^2 + (t+1)/T\epsilon$$

Therefore, we have $\|\nu_b^t - \nu^*\|_2^2 \leq \|\nu_a^t - \nu^*\|_2^2 + t/T\epsilon$, which suggests that the teacher-aware learner can always converge no slower than the naive learner up to an $\epsilon$ factor. $\qquad \square$

## B  Detailed Experiment Settings

We used two types of loss functions in all the experiment. For regression tasks, our loss function is

$$\min_{\omega \in \mathbb{R}^d, b \in \mathbb{R}} \frac{1}{n} \sum_{i=1}^n \frac{1}{2} (\omega^T x_i + b - y_i)^2 + \frac{\lambda}{2} \|\omega\|_2^2$$

For classification tasks, our loss function is

$$\min_{\omega \in \mathbb{R}^d, b \in \mathbb{R}} \frac{1}{n} \sum_{i=1}^n \sum_{k=1}^K -\mathbf{1}(y_i = k) \log p_{ik} + \frac{\lambda}{2} \|\omega\|_2^2$$

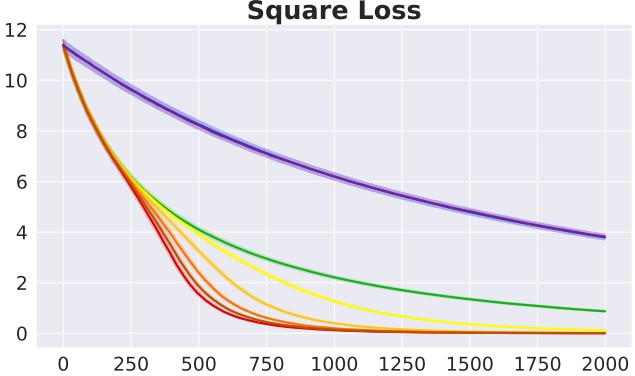

Figure 1: Square loss of the linear regression.

$$p_{ik} = \frac{\exp(\omega_k^T x_i + b_k)}{\sum_{k'=1}^{K} \exp(\omega_{k'}^T x_i + b'_k)}$$

where $\omega \in \mathbb{R}^{K \times d}$ and $\omega_k$ is the $k$-th row of $\omega$, $b \in \mathbb{R}^K$ and $b_k$ is the $k$-th element of $b$. The norm is Frobenius norm. In both regression and classification tasks, we refer to $[\omega, b]$ as $\omega^*$. In all the following experiments, we used a constant learning rate $10^{-3}$ for all the algorithms. The size of the minibatch was set to 20. As the gradient scale is different in different experiments, we used different $\beta$s. We chose the hyperparameter $\beta$ so that at the beginning of the learning, the data in the mini-batch with the smallest teaching volume has above $80\%$ probability of being selected. In the supplementary material, we show additional results of our experiments. All plots are consistent with the results in the main text. All of our experiments were run on machines with 16 I9-9900K cores and 64GiB RAM. The longest setting is the Tiny ImageNet classification, takes about 12 hours to finish (2000 iterations for 8 methods and 20 random seeds).

## B.1 Linear Models on Synthesized Data

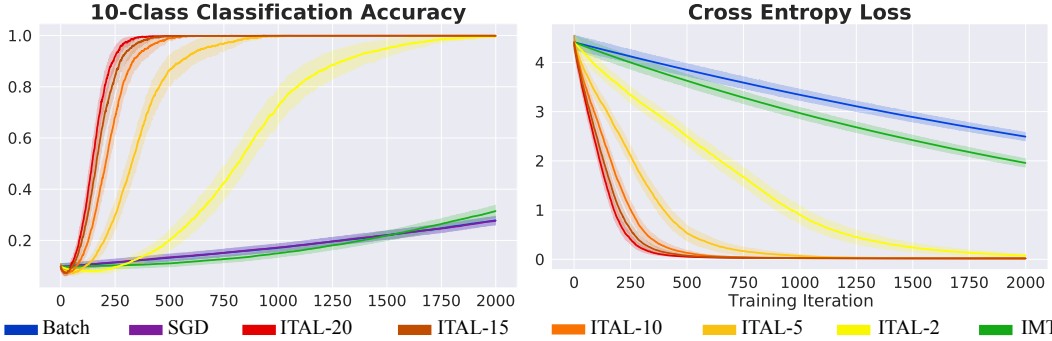

Figure 2: Classification accuracy and Cross Entropy loss of the 10-class Gaussian data classification.

For the regression task, both $\omega^*$ and $X$ are randomly generated from a uniform distribution, namely $\omega_i, b, X_{ij} \sim U[-1, 1]$. $Y = X\omega^*$. The data points have dimension 100, and $\beta$ is chosen to be 2000. For the classification task, we first randomly generate $K$ points as the center of each class from $U[-1, 1]$. Then, we use these points as the centers of Normal distributions with $\Sigma = 0.5I_{(d+1)}$. $N/K$ points are sampled from each distribution as the data. We get $\omega^*$ using the logistic regression model in Scikit-learn [4]. For classification task with 30D data, we use $\beta = 60000$. We used $\lambda = 0$ for both tasks. For the scenario of different feature spaces, we use a random orthogonal projection matrix to generate the teacher's feature space from the student's. $\omega^*$ and $\nu^*$ are multiplied with the inverse of the projection matrix to preserve the inner product. Figure 1 shows the square loss of the linear regression task. Figure 2 shows the classification accuracy and the Cross Entropy loss of 10-class classification tasks.

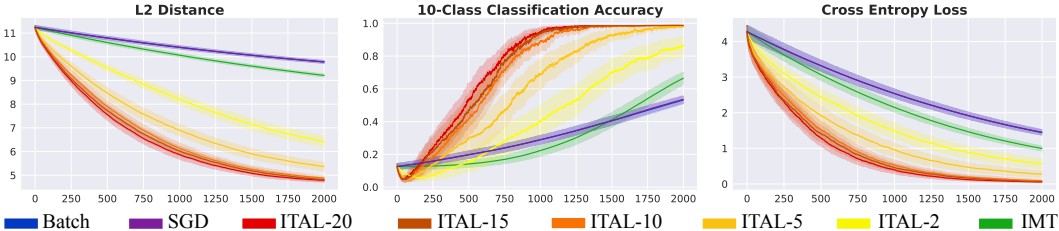

Figure 3: L2 distance, accuracy and Cross Entropy loss of the 10-class MNIST classification, in which the teacher uses 20D features.

## B.2 Linear Classifiers on MNIST Dataset

Table 1: MNIST CNN structure

|  | 20-Dim CNN | 24-Dim CNN | 30-Dim CNN |
|---|---|---|---|
| Conv 1 | 1 layer, 64 [3×3] filters, leaky ReLU | | |
| Pool | 2×2 Max with Stride 2 | | |
| Conv 2 | 1 layers, 32 [3×3] filters, leaky ReLU | | |
| Pool | 2×2 Max with Stride 2 | | |
| Conv 3 | 1 layer, 32 [3×3] filters, leaky ReLU | | |
| FC | 20, tanh | 24, tanh | 30, tanh |

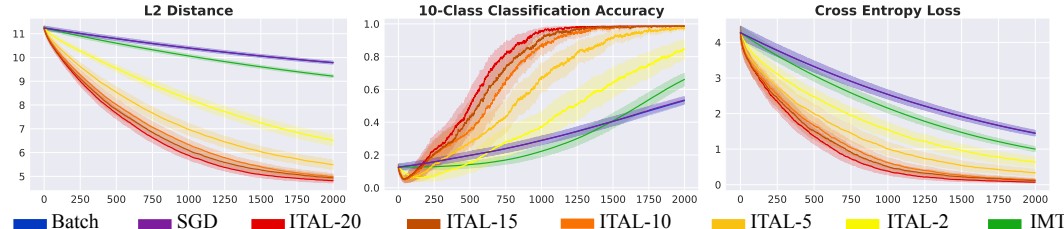

Figure 4: L2 distance, classification accuracy and Cross Entropy loss of the 10-class MNIST classification, in which the teacher uses 30D features.

We also did experiment with the MNIST dataset, which didn't mentioned in the main text for space sake. For our 10-class MNIST experiment, we trained 3 different CNNs with the similar architecture, only differing in the number of units in the last fully connected (FC) layer. The structure is summarized in table 1. All three CNNs were able to achieve above 97% test accuracy. To test our ITAL method, we had the teacher teach the parameters of the FC layer to the student. The input of this layer is used as feature vectors of the images. The learner always used features with 24D, but the teacher varied with 20D and 30D, results presented in figure. In both settings, $\beta$ is set to 30000. The FC layer weights trained with supervise learning were used as $\nu^*$. Figure 3 and 4 show the classification accuracy and Cross Entropy loss of the training.

## B.3 Linear Classifiers on CIFAR-10

The overall design of this experiment resembles the MNIST classification. We used CIFAR-10, a dataset with more enriched and complicated natural images. We trained three different CNNs with 6, 9, and 12 convoluted layers on an augmented CIFAR dataset. With 40 epochs and an adaptive learning rate, we were able to achieve about 82 percent test accuracy for all three architectures. Table 2 summarizes the CNN structure we used. To stabilize training, we used an exponential decaying $\beta$, $\beta_t = 50000(1 - 5e^{-6})^t$. We think that because the feature representation is quite different between the teacher and the student, as the iterative learning goes, the approximation error might accumulate. Thus, the learner's estimation of the teacher's data selection will be less accurate towards the end of the learning, especially for the most ambiguous examples (images prone to mistakes). At this time, using a large $\beta$ can be unstable. In other words, it is hard for the learner to reason about the teacher at

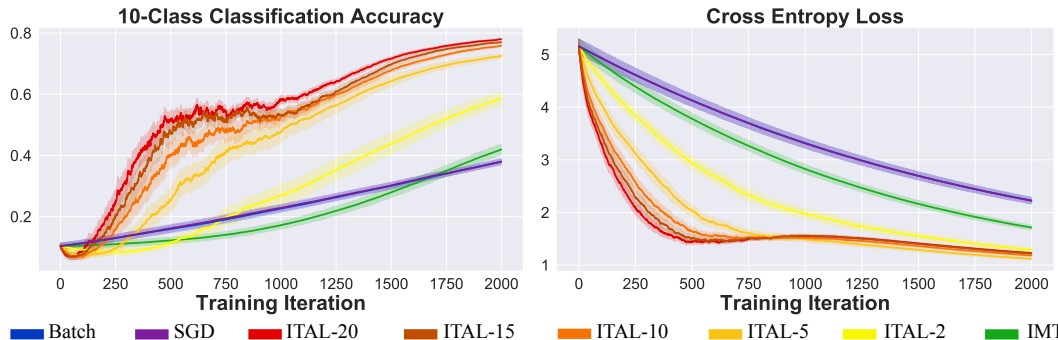

Figure 5: Accuracy and Cross Entropy loss of the 10-class CIFAR-10 classification, in which the teacher uses features extracted from CNN-6 detailed in table 2. The L2 loss curves we included in the main text section 5 figure 1c was from this setting.

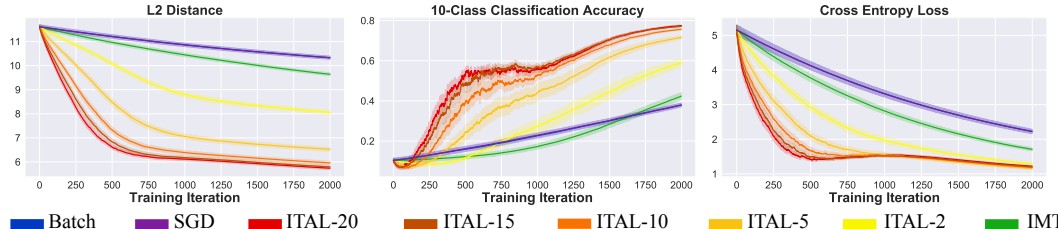

Figure 6: L2 distance, classification accuracy and Cross Entropy loss of the 10-class CIFAR-10 classification, in which the teacher uses features extracted from the CNN-12 detailed in table 2.

the end of the learning, so he should be less confident using the pragmatic information suggested by the teacher's intention. Figure 5 and 6 show the accuracy and Cross Entropy loss of this task.

## B.4 Linear Classifiers on Tiny ImageNet

The overall design of this experiment resembles the MNIST and CIFAR-10 classification. We used Tiny ImageNet, a large scale dataset with natural images. We first extracted 2048D features from VGG-13/16/19 without finetuning and then downsampled the features to 10D with a multilayer perceptron with three FC-ReLU-layers (500, 250, 10) trained with Cross Entropy loss. Figure 7 and 8 show the accuracy and Cross Entropy loss of this task.

## B.5 Linear Regression for Equation Simplification

In this experiment, we let the teacher teach a value function to the student so that he can use this value function to simplify polynomials given predefined operations. We first created an Equation Simplification dataset. we randomly generate two fourth-degree polynomials with three variables $x, y, z$ as the left- and right-hand sides of the equations. The coefficients of the polynomials are

Table 2: CIFAR-10 CNN structures.

|  | CNN-6 | CNN-9 | CNN-12 |
|---|---|---|---|
| Conv 1 | 2 layers of 16 [3×3] filters | 3 layers of 16 [3×3] filters | 4 layers of 16 [3×3] filters |
| Pool | 2×2 Max with Stride 2 | | |
| Conv 2 | 2 layers of 32 [3×3] filters | 3 layers of 32 [3×3] filters | 4 layers of 32 [3×3] filters |
| Pool | 2×2 Max with Stride 2 | | |
| Conv 3 | 2 layers of 64 [3×3] filters | 3 layers of 64 [3×3] filters | 4 layers of 64 [3×3] filters |
| Pool | 2×2 Max with Stride 2 | | |
| FC | 32 | 32 | 32 |

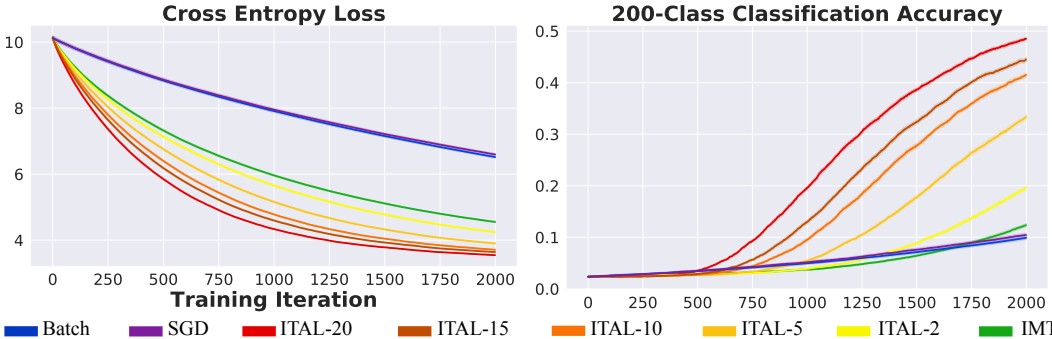

Figure 7: Top-5 accuracy and Cross Entropy loss of the 200-class Tiny ImageNet classification, in which the teacher uses features extracted from VGG-13. The L2 loss curves we included in the main text section 5 fig. 1d was from this setting.

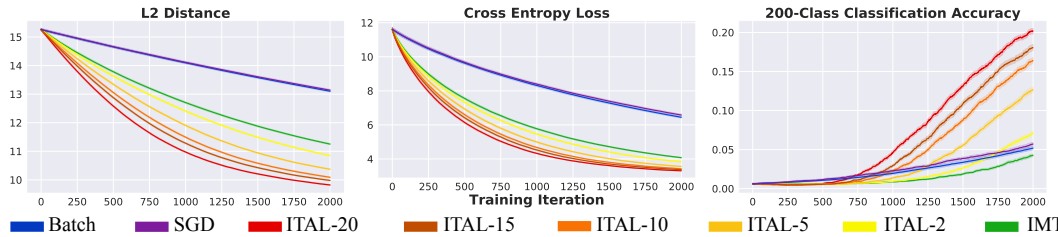

Figure 8: L2 distance, top-5 accuracy and Cross Entropy loss of the 200-class Tiny ImageNet classification, in which the teacher uses features extracted from the VGG-19.

either fractions or integers. The range of magnitude is 20 and 5 for the nominator and denominator, respectively. We define a set of operations that can be performed on an equation.

- **Scale**: scale a term by an integer factor.
- **Reduction**: reduce the fraction coefficient of a term to the simplest form.
- **Cancel common factors**: divide coefficients of all terms by the greatest common factor of integer coefficients and the nominators of fractional coefficients.
- **Move**: move a term to a specified position in the equation.
- **Merge**: merge two terms that contain the same denominators and variables with the same degrees.
- **Cancel denominators**: multiply all terms by the least common multiple of the denominators of all coefficients.

To simplify an equation, we apply operations in the following way:

1. Canceling common factors
2. Merging terms with the same denominators
3. Merging terms with different denominators by scaling the terms with the least common multiple and then applying the merge operation
4. Removing fractions in the coefficients
5. Rearranging the terms by descending degrees of $x, y, z$ with the move operation

At each step, only one operation is performed on a single term (two terms for merging), and we do not move on to the next operation until the present one is no longer applicable to the current equation. After the simplification process, all the remaining terms are on the left-hand side, while the right-hand side is simply 0. We record the series of equations generated as a simplification trajectory. Some example simplification trajectories would be:

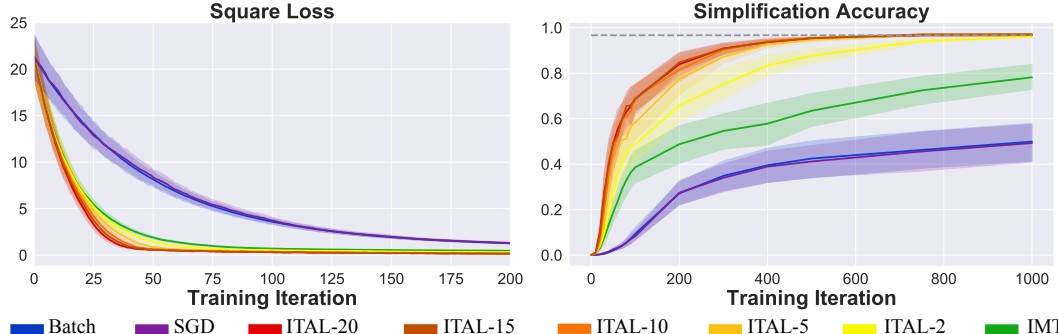

Figure 9: Square loss and simplification accuracy using the learned value function. We compared the last equation in the trajectories generated by the predefined rules and the greedy search results guided by the learned value function. Given the same teacher, teacher-aware learning algorithm outperforms naive learners in terms of accuracy and convergence rate. The gray horizontal dash line represents test accuracy using the ground truth parameter of 45D. For these results, the teacher uses 40D features, same as the L2-loss in section 5 figure 1e.

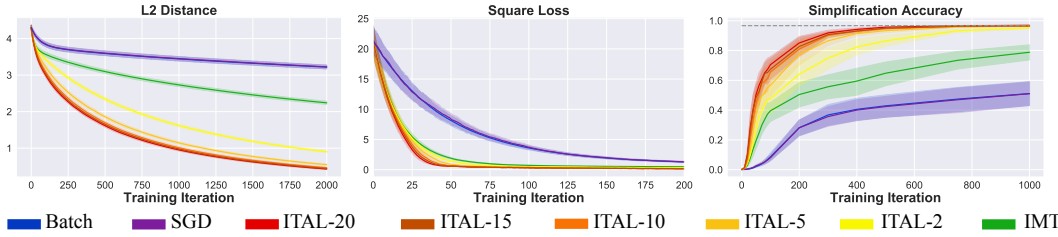

Figure 10: L2 distance, square loss and simplification accuracy using the learned value function. The teacher uses 50D features for these results.

Example equation 1 :
$$\frac{1}{4}xyz - \frac{3}{2}xyz = -14y^3 + \frac{1}{5}$$
$$-\frac{5}{4}xyz = -14y^3 + \frac{1}{5} \qquad \text{(merge)}$$
$$-25xyz = -280y^3 + 4 \qquad \text{(cancel denominators)}$$
$$-25xyz + 280y^3 = 4 \qquad \text{(move)}$$
$$-25xyz + 280y^3 - 4 = 0 \qquad \text{(move)}$$

Example equation 2 :
$$5x^3y + \frac{8}{3}z = -\frac{14}{3}z + 6xy^2z + \frac{11}{3}xz^2 + 6yz^2$$
$$5x^3y + \frac{22}{3}z = 6xy^2z + \frac{11}{3}xz^2 + 6yz^2 \qquad \text{(merge)}$$
$$15x^3y + 22z = 18xy^2z + 11xz^2 + 18yz^2 \qquad \text{(cancel denominators)}$$
$$15x^3y - 18xy^2z + 22z = 11xz^2 + 18yz^2 \qquad \text{(move)}$$
$$15x^3y - 18xy^2z - 11xz^2 + 22z = 18yz^2 \qquad \text{(move)}$$
$$15x^3y - 18xy^2z - 11xz^2 - 18yz^2 + 22z = 0 \qquad \text{(move)}$$

We applied CNN $\phi_\theta$ to learn the features of the generated equations and a linear value function wrt. these features. We first encode equations using a codebook which maps each character to a trainable vector embedding. Thus, each equation can be encoded as a matrix. Then, we treat each equation as a 3D tensor with size $1 \times W \times C$, where $W$ is the number of characters in the equation and $C$ is the length of embedding. We set $C = 30$, and $W$ ranges from 6 to 173. During training, we padded 0

Table 3: Equation CNN structure

|        | 40-Dim CNN | 45-Dim CNN | 50-Dim CNN |
|--------|------------|------------|------------|
| Conv 1 | 1 layer, 64 [5×5] filters, leaky ReLU | | |
| Conv 2 | 1 layers, 64 [5×5] filters, leaky ReLU | | |
| Pool   | 2×2 Max with Stride 2 | | |
| Conv 3 | 1 layer, 32 [3×3] filters, leaky ReLU | | |
| Conv 4 | 1 layer, 32 [3×3] filters, leaky ReLU | | |
| Pool   | 2×2 Max with Stride 2 | | |
| Conv 5 | 1 layer, 32 [3×3] filters, leaky ReLU | | |
| Conv 6 | 1 layer, 32 [3×3] filters, leaky ReLU | | |
| Pool   | 2×2 Max with Stride 2 | | |
| FC     | 40, tanh | 45, tanh | 50, tanh |

to make sure all equations in one batch form a regular tensor. We fed the encoded equations to the CNN and used the output as their feature vectors. The structure of the CNN is summarized in table 3. The value of a given equation is the inner product of its feature vector and the parameter $\omega$. The loss function is based on contrastive loss and seeks to maximize the difference between the values of the simpler equations and the complicated equations. During training, we learn the network parameters and the weight vector simultaneously with:

$$\mathcal{L}(\omega, \theta) = \frac{1}{|S_+|} \sum_{(E_i, E_j) \in S_+} \max\left(1 - \left(\phi_\theta(E_i) - \phi_\theta(E_j)\right)^T \omega, 0\right) +$$
$$\frac{1}{|S_-|} \sum_{(E_i, E_j) \in S_-} \max\left(1 - \left(\phi_\theta(E_j) - \phi_\theta(E_i)\right)^T \omega, 0\right) + \frac{\lambda}{2}\|\omega\|_2^2$$

where $S_+$ and $S_-$ hold positive and negative pairs respectively. The positive data are pairs of equations from the same simplification trajectory, where the first equation in the pair is generated after the second one. That is, the first equation is simplified from the second equation, hence having a higher value. For the negative data, we randomly select an equation from a simplification trajectory excluding the simplification result and randomly apply an operation to that equation. If the result of the operation is different from the next equation in the trajectory, we add the result-equation pair to $S_-$. Otherwise, we randomly choose a different operation until the result of the operation is not the next equation in the trajectory, and then add the pair to $S_-$. This way, we acquire pairs whose first equations have lower values than the second ones'. We train 3 sets of value functions, with the different feature dimensions, 40D, 45D, and 50D.

After we learned a value function, we utilized $\omega^*$ as the ground truth parameter. The teacher and the learner represent the equations with the learned features. The learner always used features with 45D, and the teacher used 40D or 50D, corresponding to figure 9 and 10. In all settings, $\beta$ is set to 5000. We tested the learned parameters with equations not included in the training set. Specifically, to simplify an equation, we applied all possible operations to it and obtained the outcome equation values. Then we used the greedy search to select candidates according to their values. The search ends when all the outcome equations have a lower value than the current equation. If the final equation generated by the learned value function matches with the simplification generated by our rules, we count this simplification as correct. In figure 9 and 10, we provide the square loss and the accuracy for the simplification task.

### B.6 Online Inverse Reinforcement Learning

In this experiment, we want to learn a reward function $r(s, \omega^*)$. We can define a Markov Decision Process $\langle S, A, r, P, \gamma \rangle$, where $S$ is the state space, $A$ is the action space, $r : S \to R$ is a reward function mapping from state to a real number as the reward. $P_{ss'}^a$ is the transition model that state $s$ becomes $s'$ after the agent conducts action $a$. $\gamma$ is a discount factor that ensures the convergence of the MDP over an infinite horizon. Given a reward function, using Bellman equation we have

$$V^*(s) = \max_{a \in A} \sum_{s'|s,a} P_{ss'}^a \left[r(s') + \gamma V^*(s')\right] \tag{S-6}$$

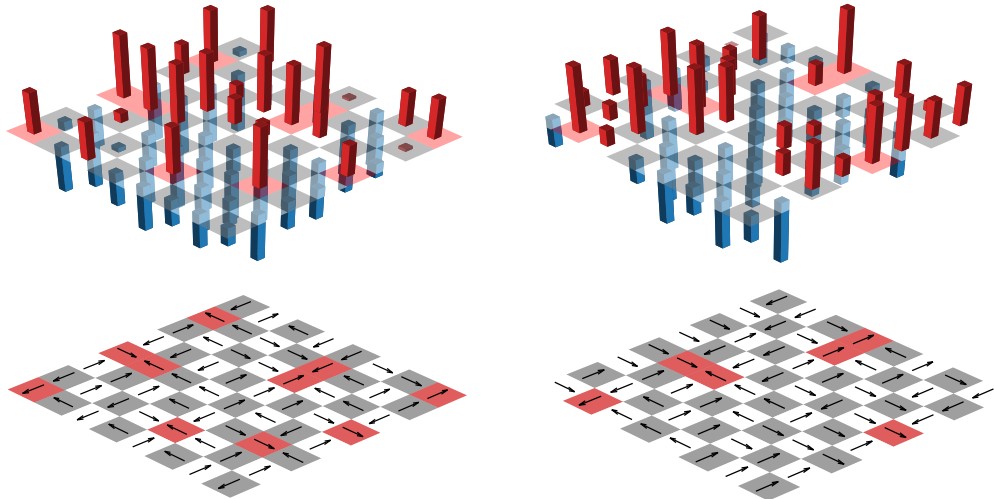

Figure 11: IRL map examples. Each map has $8 \times 8$ grids. Every grid contains a reward. Maps in the first row plots the **ground truth** rewards in each grid. Red bars represent positive rewards and blue bars represent negative rewards. The learner tries to learn a policy to walk in the map and collect the most accumulative rewards. The arrows below indicate the most probable action taken by the learner after he learned the reward function. The red grids are targets of all their neighbors.

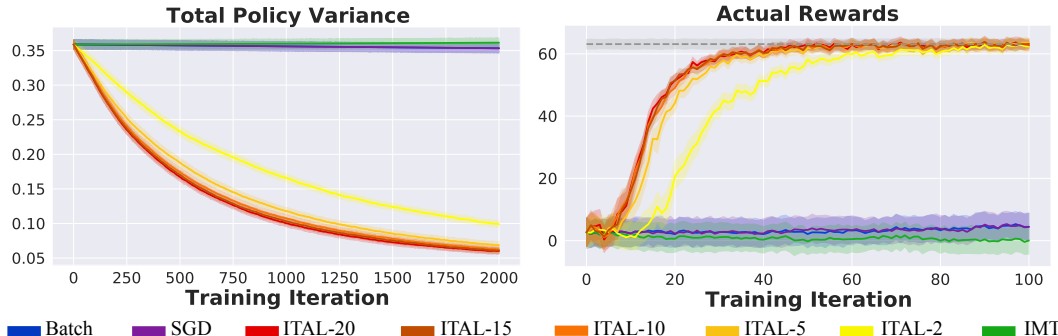

Figure 12: Total variance between the learner's policy and the teacher's policy and the actual gain of the learner during the learning process. The gray horizontal dash line represents teacher's expected accumulative reward.

$$Q^*(s, a) = \max_{a \in A} \sum_{s' \mid s, a} P_{ss'}^a \big[ r(s') + \gamma \max_{a' \in A} Q^*(s', a') \big] \tag{S-7}$$

Suppose an agent behaves by following Boltzman rationality:

$$\pi(a^t \mid s^t; \omega) = \frac{\exp\left(\alpha Q^*(s^t, a^t; \omega)\right)}{\sum_{a' \in A} \exp\left(\alpha Q^*(s^t, a'; \omega)\right)} \tag{S-8}$$

Take log-likelihood of this function we can have an objective function that the learner can optimize to learn $\omega^*$.

$$l(s^t, a^t; \omega^{t-1}) = \alpha Q^*(s^t, a^t; \omega^{t-1}) - \log \sum_{a' \in A} \alpha Q^*(s^t, a'; \omega^{t-1}) \tag{S-9}$$

$$\omega^t = \omega^{t-1} + \eta_t \frac{\partial l(s^t, a^t; \omega^{t-1})}{\partial \omega^{t-1}} \tag{S-10}$$

Then, the online IRL process can be accomodated by our learning framework. One issue is that the max operation in $Q$ is not deferentiable. Thus, we approximated max with soft-max, namely:

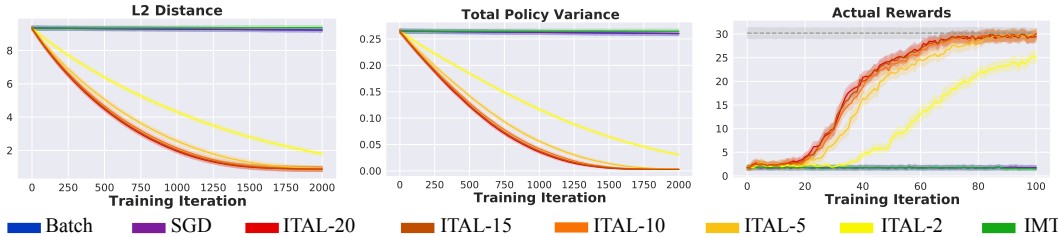

Figure 13: Learning results of sparse reward maps. All grids in the 8×8 map have 0 reward but 3 grids with reward 1. In every experiment, we randomly selected the 3 grids. Curves drawn with results using 20 different random seeds. The gray horizontal dash line represents teacher's expected accumulative reward.

$\max(a_0, ..., a_n) \approx \frac{\log(\sum_{i=0}^{n} \exp ka_i)}{n}$, with $k$ controlling the level of approximation and leveraged the online Bellman gradient iteration [3] to calculate the gradient for each step.

$$\frac{\partial V_{g,k}(s;\omega^t)}{\partial \omega^t} = \sum_{a \in A} \frac{\exp\left(kQ_{g,k}(s,a;\omega^t)\right)}{\sum_{a' \in A} \exp\left(kQ_{g,k}(s,a';\omega^t)\right)} \frac{\partial Q_{g,k}(s,a;\omega^t)}{\partial \omega^t} \tag{S-11}$$

$$\frac{\partial Q_{g,k}(s,a;\omega^t)}{\partial \omega^t} = \sum_{s'|s,a} P_{ss'}^a \left(\frac{\partial r(s';\omega^t)}{\partial \omega^t} + \gamma \frac{\partial V_{g,k}(s';\omega^t)}{\partial \omega^t}\right) \tag{S-12}$$

In every round, we randomly sample 20 $(s, a)$ pairs from $|S| \times |A|$ state-action pairs as our minibatch. Then the teacher will conduct Bellman gradient iteration. The learner will return his reward estimation for each grid to the teacher.

We used an $8 \times 8$ grid map as the environment, and the action space $A$ includes four actions up, down, left, right. See figure 11 for map examples. $80\%$ of the time, the agent goes to its target, $18\%$ of the time ends up in another random neighbor grid and $2\%$ of the time dies abruptly (game ends). We set $\gamma = 0.5$. The reward in each grid is randomly sampled from a uniform distribution, $U[-2, 2]$. If we encode each grid with a one-hot vector, then the reward parameter is a 64D vector with the $i$-th entry corresponding to the reward of the $i$-th grid. The teacher uses a shuffled map encoding as the student's. For instance, if the first grid is $[1, 0, ..., 0]$ to the learner, then it becomes $[0, ..., 0, 1, 0, ...]$ to the teacher. See figure 12 for the actual accumulative reward acquired by the agent during learning.

In addition to the environment with random dense rewards, we tested the teacher-aware learner in a sparse reward environment. Each time, we only pick 3 grids randomly to assign non-zero reward. Our algorithm still shows robust performance. Results in figure 13.

## B.7 Adversarial Teacher

Table 4: Selection of $\beta$s in the adversarial teacher experiments. For cooperative teachers, the absolute values of the $\beta$s are the same, only the signs are flipped.

| Experiment | Value of $\beta$ |
|---|---|
| Linear Classifiers on Synthesized Data | -60000 |
| Linear Regression on Synthesized Data | -5000 |
| Linear Classifiers on MNIST Dataset | -30000 |
| Linear Classifiers on CIFAR Dataset | $-50000(1 - 5e^{-6})^t$ |
| Linear Classifiers on Tiny ImageNet Dataset | $-1000$ |
| Linear Regression for Equation Simplification | -5000 |
| Online Inverse Reinforcement Learning (Random Rewards) | -25000 |
| Online Inverse Reinforcement Learning (Sparse Rewards) | -30000 |

We further test the robustness of our algorithm with an adversarial teacher, who, instead of choosing the most helpful data, chooses the least helpful one. She replace the $\arg\max$ in equation (2) in the main text with $\arg\min$. In this scenario, a naive learner can barely learn, but the teacher-aware learner still shows steady improvement. See table 4 for the $\beta$ used in these experiments.

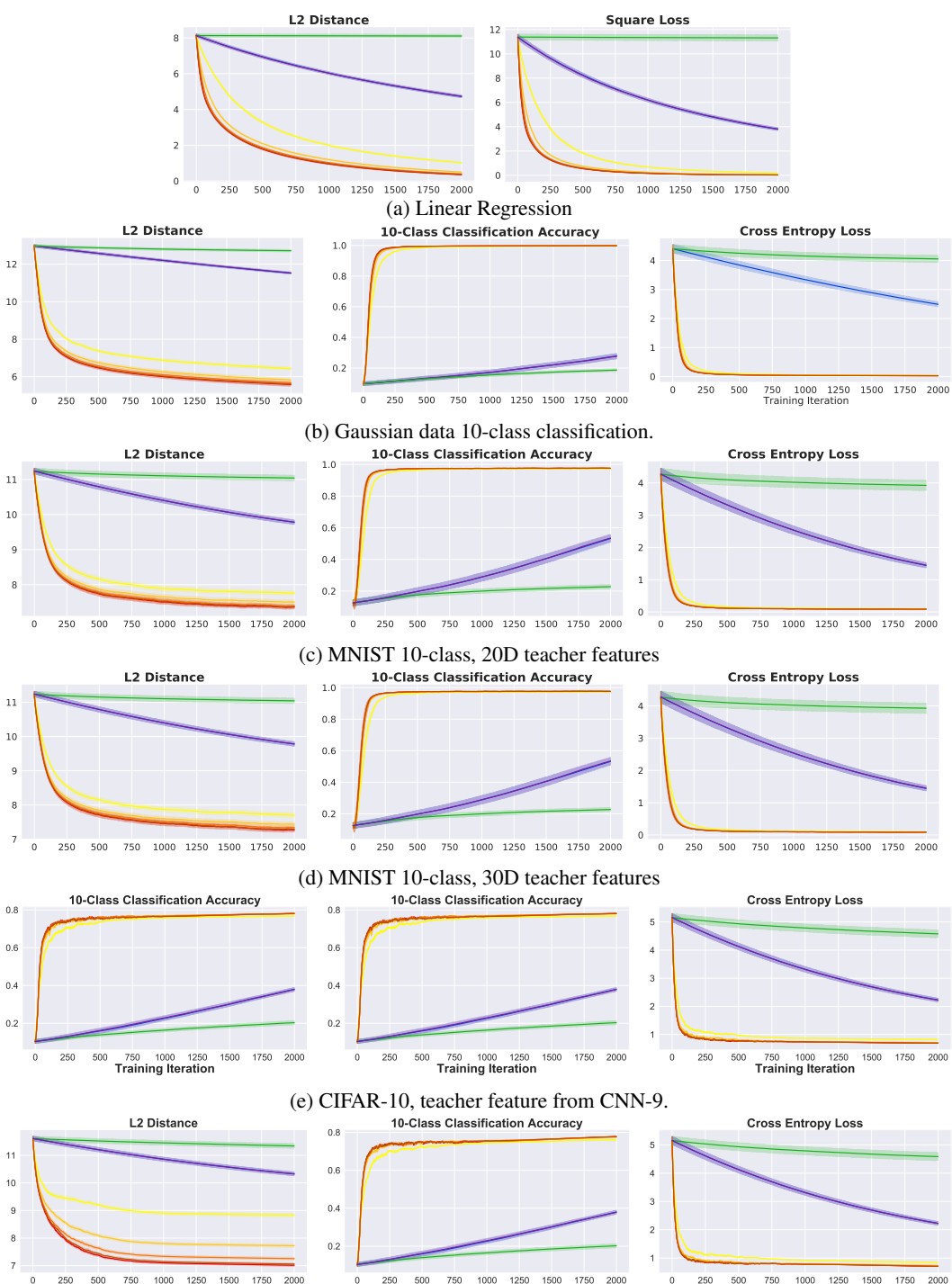

(a) Linear Regression

(b) Gaussian data 10-class classification.

(c) MNIST 10-class, 20D teacher features

(d) MNIST 10-class, 30D teacher features

(e) CIFAR-10, teacher feature from CNN-9.

(f) CIFAR-10, teacher feature from CNN-12.

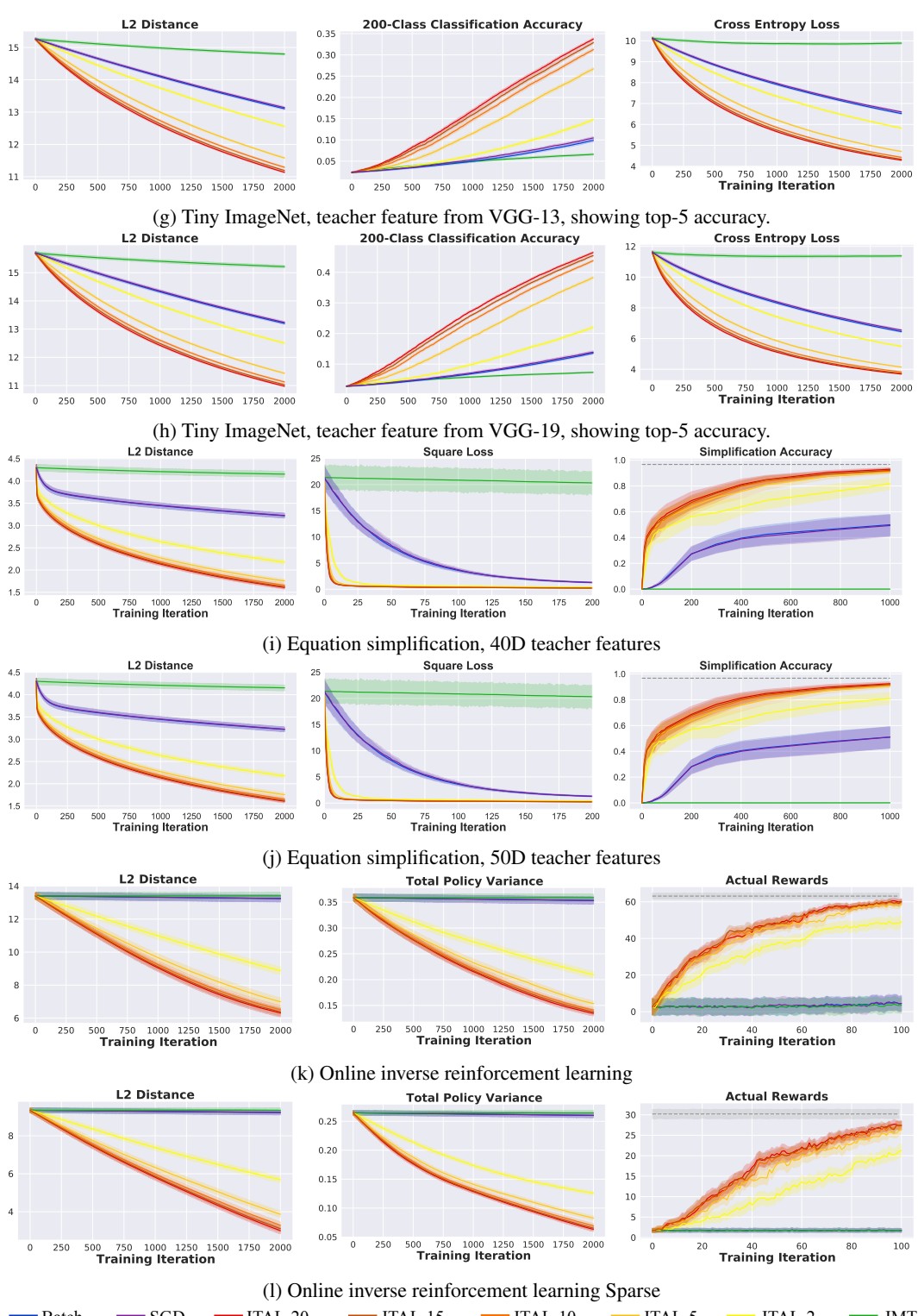

(g) Tiny ImageNet, teacher feature from VGG-13, showing top-5 accuracy.

(h) Tiny ImageNet, teacher feature from VGG-19, showing top-5 accuracy.

(i) Equation simplification, 40D teacher features

(j) Equation simplification, 50D teacher features

(k) Online inverse reinforcement learning

(l) Online inverse reinforcement learning Sparse

Figure 14: Adversarial teacher results. With an adversarial teacher, a naive learner can no longer learn effectively. ITAL still learns efficiently. SGD and batch learning are included for comparison.

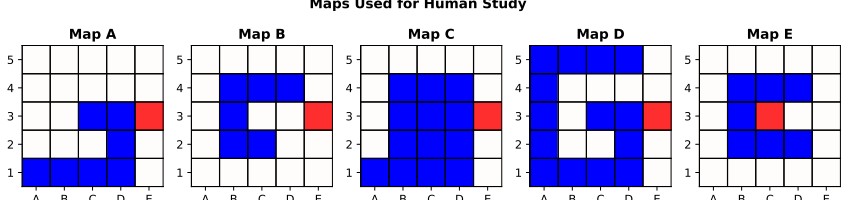

Figure 15: Map configurations

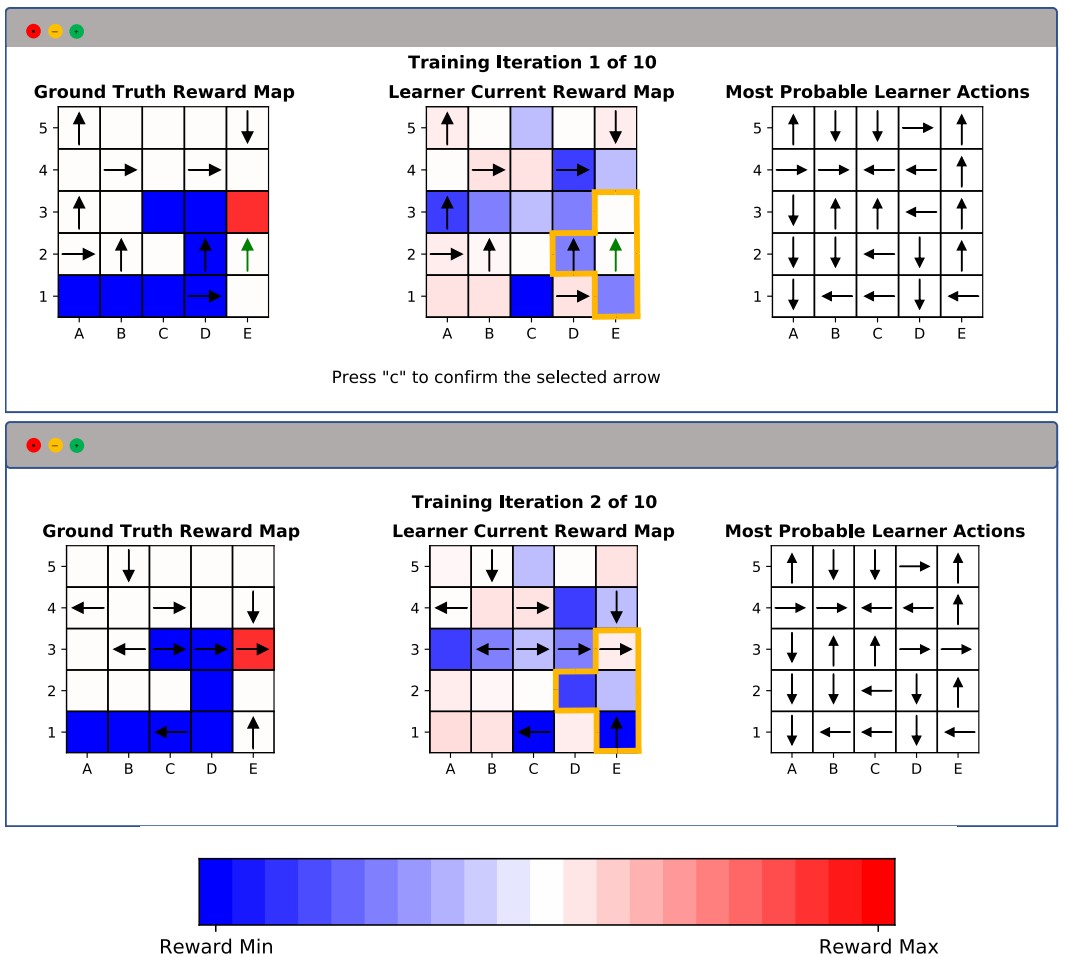

Figure 16: Actual interface used in the human study. The reward spectrum will be shown to the subjects at the experiment introduction. In this example, the subject chose the green arrow as the example at the 1-st iteration. Then, as we annotated with the orange T-shaped boxes, the estimated reward of the target tile of the green arrow increased, while rewards of the arrow source and the surrounding neighbors decreased. The orange boxes were not included in the human study.

## B.8   Human Teacher

We conducted a proof-of-concept human study on 20 university students, 10 females and 10 males. We want to validate that our teacher-aware learner can also outperform naive learners given a human teacher. In other words, our teacher model can be applied to human teachers, despite of their potentially different pedagogical patterns. The goal of the experiment is for the participant to teach the reward of a ground-truth reward map to a learner. To reduce human subjects' cognitive burden, we use three types of tiles (red, blue and white) on the map to represent bad, good and neutral grids. We

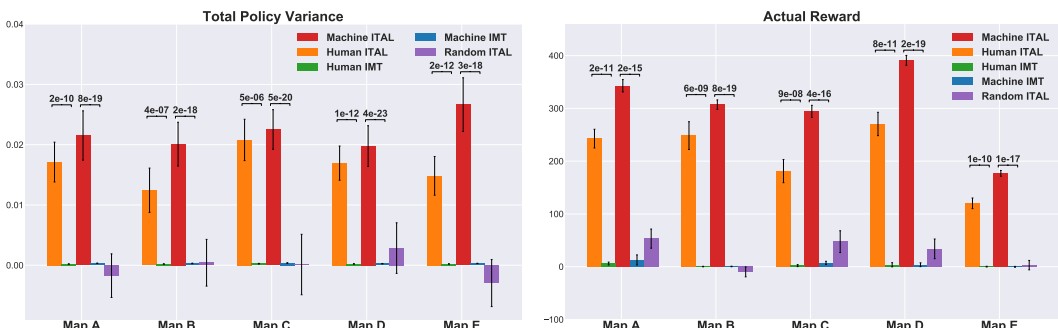

Figure 17: Human study results. All the p-values are calculated with paired t-test.

(a) Map A

(b) Map B

(c) Map C

(d) Map D

(e) Map E

Machine ITAL    Human ITAL    Random ITAL    Machine IMT    Human IMT    Teacher

Figure 18: Learning curves for each map.

used 5 different map configurations shown in figure 15. The learner's current reward map is shown to the human participants during the entire teaching session. As the reward values are continuous at the learner's side, we used the color pallet in figure 16 to render the grids in the learner's map. We also included a map indicating the most probable actions the learner will take given his current reward map, so that the human teacher can tell which grid the learner attaches a higher reward if some neighboring grids have similar colors. The directions of these arrows are calculated with value iteration using the learner's current reward parameters. An example human interface was shown in figure 16. In each time step, ten arrows will be drawn on ten randomly sampled grids. Selecting one of the arrows tells the learner that he should follow the arrow's direction if he was at this grid. Then the learner will update his reward parameters based on this instruction using the same equation (S-10) as in section B.6.

We hold the experiment as a Jupyter Notebook [2] and launch it via Binder [1]. We first introduce the experiment logic to the human subjects and include a short warm-up phase for the subjects to get familiar with the learner's update process. Then, we let the subjects to teach the maps, starting from Map A to Map E. Every subject needs to teach both a teacher-aware learner and a naive learner, whose order is randomly determined. For every map, the initialization of the two learners are the same for the same human subject and different across subjects. Like the inverse RL experiment in section B.6, we evaluate the learning results in terms of L2-distance between the learners' reward parameters and the ground-truth parameters, the total variance between the learners' policy and the policy derived with the ground-truth reward and the actual accumulated reward acquired by the learner after the learning completes.

The results are presented in figure 17. The advantages of the teacher-aware learners are significant ($p$-value $< 0.01$) on all measurements, computed with a paired t-test. We also did an ablative study, in which the human teacher was replaced by a random teacher. As shown in the figure 17 and 18, when paired with a random teacher, the teacher-aware learner doesn't show any advantage and has much larger variance. That is to say, the teacher model only benefits the learning when it matches with the actual teacher data selection process. Otherwise, the teacher-aware learner will over-interpret the data he receives. Figure 18 shows the learning curves of all the map configurations.