# OpenReview forum: "Iterative Teacher-Aware Learning"
_NeurIPS.cc/2021/Conference — NeurIPS 2021 Poster_

### Official Review · Reviewer_LPK1 · 2021-07-16

**Rating:** 6
**Confidence:** 4

**Summary:**

The paper presents a gradient optimization-based teacher-aware learner who can incorporate a teacher's cooperative intention into its likelihood function, and learn faster in comparison to the existing naive learning algorithms for machine teaching. The authors evaluate the proposed algorithm empirically on several machine learning tasks such as classification/regression/IRL and also on a simple setting where instruction/advice is taken from human teachers. Their results show that learning is indeed faster with a teacher-aware learner.

**Limitations And Societal Impact:**

The paper is still in its nascent stages and far from deployment in the real world. Hence, negative societal impacts are not a concern just yet.

**Main Review:**

**Pros**

* The modeling of a teacher-aware learner is interesting.
* The experimental evaluation catered to different types of learning models and showed in all the diverse settings (classification/regression/IRL) that a teacher-aware learner has accelerated learning.


**Cons**

* Writing: The paper could have been made clearer, particularly equation (6) in Section 4.2. Since this is a crucial equation illustrating the gradient update rule for such a learner, it could have been better annotated, and upon annotation, refer to its different parts in the subsequent text that follows. There is a typo on line 202 where I think Eq.5 was mentioned, by it should have been Eq.3.

* Evaluation: While the evaluation was exhaustive, one could elaborate further on the Online IRL setting in the main paper, and put the details of the Linear classifier on natural image datasets in the appendix instead. The Online IRL setting is a more interesting set-up for the problem. Furthermore, the experiments with the human teacher were overly simplified for this setting. The method has the potential for dealing with a more complex instruction set of a human teacher, given that the learner model is general enough and operates under the realistic assumption of a sub-optimal teacher.



**General Comments**
The paper presents important theoretical guarantees for the proposed teacher-aware learner under a set of practical assumptions. The learner model has the potential of conducting counterfactual reasoning by comparing the teacher's selected instruction with that of others, in the setting that the teacher's instruction set is finite. The experimental evaluation also looks promising, particularly the verification with human teachers. It would be interesting to evaluate the performance of such a learner in a real-world setting and explore empirically its robustness to noise in teacher instruction.

*Originality*: Novel Approach

*Clarity*: Largely clear.

*Quality*: Good

*Significance*: High

**Time Spent Reviewing:**

3

---

> ### Author Response · Authors · 2021-08-08
> **Reply to Reviewer LPK1**
>
> * Q: Better annotate Eq. (6)
>     - A: We explain each term in Eq. (6) in lines 24-26. The first two terms in Eq. (6) are the same as the naive learner, which updates the current parameter given the teacher-selected data. The third term calculates the difference between the gradient of the selected data and the expected gradient of the whole mini-batch. This difference accommodates the context of data selection, i.e., why does the teacher choose one example instead of the others in the batch.
> * Q: Typo in line 202
>     - A: Thank you for pointing out this typo, it should indeed be Eq. (3). We’ll fix this in the final version.
> * Q: Paper organization
>     - A: We’ll adjust the organization of our paper in our final version to better emphasize more distinctive and interesting experiments. At this moment, we refer readers to our supplementary material, where we provided very thorough experimental details and additional results.

---

### Official Review · Reviewer_djzf · 2021-07-17

**Rating:** 6
**Confidence:** 3

**Summary:**

**High-Level Summary**

This paper studies cooperative teacher and learner settings. In this setting, the teacher and learner are trying to solve a classification task with input $x \in \mathcal{X}$ and label $y \in \mathcal{Y}$. The setup works in an online fashion where at each time step, the teacher selects an example $(x, y)$ to present to the student from a large dataset and the student learns from this example. The teacher knows the optimal parameters. Previous work has focused on the setting where the teacher is aware of the learner and tries to select the example to ensure the student learns the fastest. This paper studies an extension where the learner is also aware of the teacher's feedback (teacher-aware learner). The key novelty is in the learner trying to maximize its parameters to fit the feedback provided by the teacher while also trying to predict the distribution the teacher is using to sample the data from. This has an additional benefit in that it can help the learner perform "pragmatic reasoning". This is related to previous work on Rational-Speech Act models (e.g., Golland et al., 2010). The key difference is that previous work on RSA provided limited theoretical analysis and was concerned with specific tasks e.g., predicting actions while this work focuses on a more abstract task of learning parameters for classification.

**Limitations And Societal Impact:**

See above for a list of concerns some of which remark on limitations.

**Main Review:**

**Technical Summary**

The teacher and learner are aware of the input and output space, the model class and the loss function. The teacher and student receive the same input up to a fixed deterministic mapping ($x$ vs $\tilde{x}$) and have a different mapping of parameters ($\nu$ vs $\omega$). The teacher has access to its optimal weight function $\omega^\star$, a dataset to pick examples from at time step $t$, and assumes that the learner is performing gradient descent on parameters wrt the loss function. The teacher picks an example such that a single step of gradient descent would bring the teacher model closest to the optimal values.  This is re-written in terms of the student model and is approximated by assuming that the student tells the teacher of the inner product between *every* data point in the dataset and its current parameter. In return, the student model does two things. It first updates its parameters to fit the example $(x, y)$ chosen by the teacher. However, it also updates it parameters to predict the distribution used by the teacher to chose the dataset. These two models use the same parameters. A subtle difference is that the teacher's model uses a point-mass distribution while the student approximates it using soft energy-based distribution.
A result is presented showing that the proposed approach under some assumption does not worse than a naive learner where the student just do gradient descent without any pragmatic reasoning (but the teacher still selects samples carefully).

**Questions and Concerns**

1. I have some difficulty understanding the setup. Firstly, is there a deterministic mapping between $\omega$ and $\nu$ similar to $x$ and $\tilde{x}$. Further, it is assumed that $\langle x, \omega \rangle $ = $\langle G(x), \nu \rangle $. If yes, then how is this $\nu$ related to $\omega$. This assumption seems to be necessary for the theory to work. E.g., in going from equation 2 to 3. Lastly, how is $\omega$ evolving with time? Is the teacher performing this update or the world? Is $\omega$ affected by student's progress?

2. It is said that the teacher know's that student is following equation 1 (line 177). But then it is stated that the teacher does not the learning mechanism (line 176).

3. If the teacher assumes that the student is following equation 1 but then the student does pragmatic reasoning then it breaks down this assumption. Shouldn't an infinite recursion occur that should be analyzed?

4. The teacher has access to the dot product of every data point with the student's current parameter (line 200-203). This sounds quite strong and would increase the communication complexity of learning significantly. Further, the teacher even knows the learning rate scheduling of the learner. In real life, the learner might only appear as a black box with limited APIs.

6. In my understanding, theoretical guarantees only show that under certain assumptions the proposal cannot do worse than a naive learner and does not demonstrate the superiority of the proposed approach. This is helpful but it also shows that the impact of novelty is primarily demonstrated via experiments. I'll request authors to clarify this point if my understanding is incorrect.

**Presentation**

1. Equation 4 has integration over the dataset which seems incoherent. I believe the authors meant summation.

2. Perhaps consider using teacher and student instead of teacher and learner for reasons of symmetry.

**Time Spent Reviewing:**

3

---

> ### Author Response · Authors · 2021-08-08
> **Reply to Reviewer djzf**
>
> * Q: “is there a deterministic mapping between ω and ν similar to x and x~. Further, it is assumed that ⟨x,ω⟩= ⟨G(x),ν⟩. If yes, then how is this ν related to ω. “
>     - A: In (Liu et al. 2017), going from eq. 2 to 3 does require the same feature representation, i.e. $x=\tilde{x}$ and $\omega=\nu$. With different representations, the teacher keeps an estimation of the learner’s parameter in her mind and updates it with her feedback to approximately maintain equal inner products. In practice, $G$ may not have a closed-form. One possible $G$ with explicit form is to generate the student’s feature space by applying an orthogonal projection to the teacher’s feature space.
>
>     - In our paper, we don’t assume, $\langle x, \omega^* \rangle = \langle \tilde{x}, \nu^* \rangle$. We relax it with a less constrained assumption in Theorem 1: the data selected by the teacher also has the largest teaching volume calculated with the student’s feature representation. In other words, the feature representation doesn’t need to have equal inner products, but only needs the best example to match between the teacher and the learner.
>
> * Q: “how is ω evolving with time? Is the teacher performing this update or the world? Is ω affected by student's progress?”
>     - A: In our paper, we have $\omega$ fixed across time and student behavior.
>
> * Q. It is said that the teacher know's that student is following equation 1 (line 177). But then it is stated that the teacher does not the learning mechanism (line 176).
>     - A: In lines 176-177 we mean that the teacher thinks the learner follows the naive learning algorithm defined in eq. 1 but she doesn’t know the actual learning mechanism of the learner.
>
> * Q: If the teacher assumes that the student is following equation 1 but then the student does pragmatic reasoning then it breaks down this assumption. Shouldn't an infinite recursion occur that should be analyzed?
>     - A: Theoretically, mutual reasoning in multiagent systems can continue infinitely or until convergence. In practice, to avoid intractability, we need to stop at a certain recursion level. As we mentioned in lines 172 and 173, our recursion level (level-1 helpful teacher with level-2 teacher-aware learner) was used in previous works and proved to match with human cognitive capability.
>
> * Q: The teacher has access to the dot product of every data point with the student's current parameter (line 200-203). This sounds quite strong and would increase the communication complexity of learning significantly.
>     - A: First, as we assume the teacher only has minimal prior knowledge about the learner’s current status, i.e. $\nu^t$, we let the teacher query the student with the current data set and collect the inner product as the only way to estimate the student’s current status. The same query and feedback design was also used in (Liu et al. 2017).
>     - Second, using inner product supports the teacher and the learner have different feature representations. Because the task, i.e. the loss function $l(\cdot)$ and the model $h(\cdot)$, is common knowledge of the agents, knowing the inner product, the teacher can still calculate the learner's loss for training data even if they have different feature representations of the data.
>     - Third, communicating inner products is much more efficient than sharing parameters, as the size of the mini-batch in each iter is usually much smaller than the dimension of the parameter. In addition, when agents have different feature representations, shared parameters are indecipherable.
>
> * Q: Further, the teacher even knows the learning rate scheduling of the learner. In real life, the learner might only appear as a black box with limited APIs.
>     - A: We agree that the learning rate may not always be available in practice. However, cooperation among agents requires estimation of a partner's behavior model and this estimation doesn’t need to be precise for the cooperation to be successful. Our human teacher experiment proved that ITAL improves learning even when the teacher doesn’t know the learning rate exactly. In this paper, we focus on improving learning when the characteristics of the teacher and the learner are known but the status ($\omega^*, \nu^t$) is unknown. In our future work, we’ll study the case when characteristics are also unknown. These may need multiple rounds of interactions (teach several different $\omega^*$) between the teacher and the learner.
>
> * Q: In my understanding, theoretical guarantees only show that under certain assumptions the proposal cannot do worse than a naive learner and does not demonstrate the superiority of the proposed approach. This is helpful but it also shows that the impact of novelty is primarily demonstrated via experiments. I'll request authors to clarify this point if my understanding is incorrect.
>     - A: Since our experiment results have already shown that our method outperforms existing methods, we do not try to prove that our method strictly improves existing methods. However, we believe that such an improvement can be proved with additional assumptions on the dataset, and we leave it as future work.

---

> > ### Comment · Reviewer_djzf · 2021-09-01
> > **Update**
> >
> > Thank you for your timely response. Based on the response, here are some comments:
> >
> > 1. **Readability:** I think the paper seems to depend a bit heavily on Liu et al, 2017 and it would do justice to the paper to separate this out in the background. E.g., the paper can benefit significantly from details of how is $G$ chosen, clarification that $\omega$ is fixed, etc. Space for this can be saved, for example, by using paragraph format (\paragraph{...}) instead of subsections in the experiment section (e.g., line 318 takes a lot of white space). You may also want to state very clearly what does the teacher know, what does the student know, and what are the interface of communication. Pseudocode can be very helpful.
> >
> >
> > 2. From this: _"in lines 176-177 we mean that the teacher thinks the learner follows the naive learning algorithm defined in eq. 1 but she doesn’t know the actual learning mechanism of the learner."_, my understanding is that the learner is not necessarily doing gradient descent but can be doing any optimization. The teacher assumes the learner is doing gradient descent but this may not be true. Is this correct?
> >
> >
> > 3. I agree that inner-product sharing is more efficient than parameters. Thanks for the clarification on this one. However, another concern I have over here is that inner-product don't seem that natural object to transfer. More natural feedback maybe bandit feedback (e.g., tell if the inner product of one term is greater than other, etc). You may want to consider this for future work. I won't use this point negatively in my score.
> >
> >
> > 4. I, honestly, think the theory part here is not that interesting since only a guarantee of _"no worse than naive"_ is proven. If authors find it easy to prove an improvement under some mild assumptions, then I highly encourage them to do so. My main takeaway here is mostly the experiments.
> >
> >
> > Overall, I am concerned with readability here. I would request the authors to clarify point 2 above. I'll re-read the paper and Appendix, but for now, I'll reduce my confidence.

---

> > > ### Author Response · Authors · 2021-09-02
> > > **Thank you for your feedback.**
> > >
> > > We really appreciate the additional suggestions and comments. We will address them as follows.
> > >
> > > ---
> > > **Q1:** Readability
> > >
> > > - **A1:** Thank you very much for the readability improvement suggestions, we’ll accommodate them in the final version to better arrange our paper. Also, we included pseudocode in our supplementary. We’ll consider moving it to the main text if it can better deliver our algorithm and the space permits.
> > >
> > > ---
> > > **Q2:** “my understanding is that the learner is not necessarily doing gradient descent but can be doing any optimization. The teacher assumes the learner is doing gradient descent but this may not be true.”
> > >
> > > - **A2:** Your understanding is correct. In our paper, the naive learner is using gradient descent to learn the concept, which is currently the most common and widely used optimization algorithm for machine learning. Therefore, for our proposed teacher-aware learner, it also uses gradient descent to learn the concept, but with a modified loss function (defined in Eq. (6) in our paper) to accommodate the teacher’s cooperation. Even if the student doesn’t strictly follow the teacher’s assumption in optimization (e.g., using other optimization algorithms rather than gradient descent), as long as the student is trying to minimize the modified loss function in Eq. (6), the cooperation can still lead to meaningful learning outcomes.
> > >
> > > ---
> > > **Q3:** “However, another concern I have over here is that inner-product don't seem that natural object to transfer. ”
> > > - **A3**: Thank you very much for your suggestions about using bandit feedback. In fact, in order to transfer inner-product, we also only need to transfer a single real number, instead of two vectors. In this sense, both the inner product and the bandit feedback suggested by you are natural. In addition, the inner product naturally appears in many machine learning models such as linear models and kernel methods, because the loss functions for these models only depend on the inner product between the inputs and the model’s parameter.  Furthermore, given convex loss functions in our setting, the inner product provides additional optimization simplicity. In our future work, we will consider the bandit feedback as well.
> > > ---
> > >
> > >
> > > **Q4:** “ the theory part here is not that interesting since only a guarantee of "no worse than naive" is proven”
> > >
> > > - **A4:** As our work is the first to propose teacher-aware learner for parameter learning, we aim to deliver a similar guarantee as existing machine teaching literature, namely a “no worse than” guarantee, under minimal assumptions in our settings. If more assumptions are being made, we can deliver stronger guarantees in the sense of “strictly better than”. Yet we need to trade additional assumptions for this stronger guarantee. So there is a tradeoff between the assumptions being made and the strength of the theoretical guarantee. We will comment on this point in the discussion and conclusion section.

---

> > > > ### Comment · Reviewer_djzf · 2021-09-05
> > > > **Thank you for response. Naturalness and Connection to Human Learning seem overstated**
> > > >
> > > > Thanks for your response. Comments on your response:
> > > >
> > > > 1. Readability: I think supplementary material has some very good details. It is obvious that the authors have put in lots of effort into this paper. I appreciate that! I highly recommend moving Algorithm 1 into the main paper. There is a typo in Algorithm 1 on line 4 (it should be $D^t$ instead of $D$).
> > > >
> > > >
> > > > 2. Thanks for the confirmation. But the paper does not always say this clearly. In fact, Algorithm 1 counteracts this assumption by clearly stating that the learner is doing gradient descent. Consider defining an abstract optimization algorithm `optimizer` and make the student learner use that instead of gradient descent. Algorithm 1 should then call this routine instead of showing gradient descent. Under some conditions on `optimizer`, one maybe able to show similar guarantees.
> > > >
> > > >
> > > > 3. Naturalness doesn't equate to communication complexity. While both bandit feedback and inner product are scalar valued, the former is more natural in my view. E.g., in a bandit setting, a user may be shown two apples and asked to pick the better one. In the inner-product term, the users are asked to report some scores which seem less natural than picking an item. Also, I don't think the code in Algorithm 1 is how human teaching works, or at least there is a significant jump in reasoning there.
> > > >
> > > >
> > > > 4. I understand the tradeoff between assumptions and guarantees. I personally, tend to find "no worse" guarantee as not at all interesting. This may be the norm but I don't find that as a good reason. In fact, the more interesting question would be: what is the minimal set of assumptions under which we have improvements. This can be done by stating assumptions and showing improvement via a theorem, and then showing via lower bounds that these assumptions are minimal.
> > > >
> > > >
> > > > Overall, I think there are interesting qualitative results here. The figures in the appendix look really impressive. However, the paper needs upliftment in readability, a playing down of connection to human teaching, and perhaps an attempt at finding the minimal set of assumptions for improvement. I believe that the authors can fix the paper on point 1 in time for camera-ready but I am less certain about the other two. I won't be disappointed if this paper is accepted, but I don't find myself giving a score higher than 6 (weak accept). I do want to state clearly that I don't find anything objectively wrong with the core algorithm or issues with the proof.

---

### Official Review · Reviewer_WHuF · 2021-07-20

**Rating:** 6
**Confidence:** 4

**Summary:**

The manuscript presents iterative teacher aware learning, an approach in which learners learn from teaching while incorporating the intent of the teacher for continuously parameterized models.

**Ethical Concerns:**

The response to 5b is a bit curious. The authors report empirical results with human teachers, which would suggest they ran an experiment. I assume they obtained IRB approval. However, the response to question 5b is no? (Even when we think there is no harm, the IRB must be consulted to receive a determination.

**Limitations And Societal Impact:**

No discussion of limitations or societal impact were included.

**Main Review:**

The paper presents iterative teacher aware learning (ITAL) which allows teaching in continuous spaces, and learners who take advantage of the teaching intent to augment learning. The approach builds on prior work in machine learning and cognitive science, and is distinctive in incorporating two features: learners who learn from teaching *and* the teacher's intent, and the ability to learn in continuous parameter spaces. Prior research has done each of these, but not together, which represents the core novel contribution of the paper. Update rules are derived that extend gradient descent methods to incorporate the teacher's selection process. Theoretical results prove that teacher aware learners learn no slower than non teacher aware. Empirical results confirm theoretical analysis for an array of problems including learning from human teaching.

Positives:
- The extension to learning from cooperative teaching in continuous parameter spaces is interesting.
- The theoretical results are clear and relevant.
- The experiments show ITAL learns faster than iterative machine teaching (IMT).

Weaknesses:
-  The heuristic nature of the teacher model was not analyzed or scrutinized, which limited possible insights provided by the paper.
- The learner feedback as an inner product is a curious choice that was not well motivated or explained.
- The connection to human learning is limited and over stated in some places.

Detailed comments:
- "diametrically distinctive" Please check the word usage here.
- "the learning mechanism" What is the learning mechanism?
- " Yet, their findings cannot be directly applied to more practical teaching scenarios" Why? Please explain.
- "Thus, we keep leveraging the greedy heuristic to model our cooperative teacher and generalize it to a non-omniscient teacher who doesn’t fully know the learner in every iteration" I do not understand the logic here.
- "This can be done approximately if, in every round, the learner gives the inner products of ν t−1 and the data to the teacher as feedback." What does this mean intuitively?
- " It has been shown that cooperative teachers using Eq. (5) can substantially " Do you mean equation 3?
- " to better approximate ν∗, in practice, we plug" What is the justification for this?
- Figure 1 is too small to read.
- "Pedagogy has a profound cognitive science background, but it receives limited attention in recent machine learning works" This isn't quite accurate. Earlier in the paper a number of machine learning papers were cited.

~~~~~~
Based on the reviewing discussions, I have decremented my rating to a 6.

**Time Spent Reviewing:**

4

---

> ### Author Response · Authors · 2021-08-08
> **Reply to Reviewer WHuF**
>
> * Q: “The heuristic nature of the teacher model was not analyzed or scrutinized, which limited possible insights provided by the paper.”
>     - A: We are using a greedy heuristic for the helpful teacher. As mentioned in lines 185-189, the merits and insights of this heuristic teaching method were introduced in (Liu et al. 2017). In equation (2), the first term represents the difficulty of an example, while the second term represents the usefulness of an example. Due to the space limit, we refer readers to (Liu et al. 2017) for more details. We believe better heuristics exist given more specific questions and limited scope, but for a generic parameter learning problem setting that allows different feature representations between agents, such a greedy heuristic is still the most practical way of teaching.
>
> * Q: The learner feedback as an inner product is a curious choice that was not well motivated or explained.
>     - A: In the paper, learning is modeled as a multiagent communication process. The inner product serves as a good way to model it in the following aspects. First, as we assume the teacher only has minimal prior knowledge about the learner’s current status, i.e. $\nu^t$, we let the teacher query the student with the current data set and collect the inner product as the only way to estimate the student’s current status. The same query and feedback design was also used in (Liu et al. 2017).
>
>     - Second, using inner product supports the teacher and the learner have different feature representations. Because the task, i.e. $l$ and $h$, is common knowledge of the agents, knowing the inner product, the teacher can still calculate the learner's loss for training data even if they have different feature representations of the data.
>
>     - Third, communicating inner products is much more efficient than sharing parameters, as the size of the mini-batch in each iteration is usually much smaller than the dimension of the parameter. In addition, when agents have different feature representations, shared parameters are indecipherable.
>
> * Q: The connection to human learning is limited and over stated in some places.
>     - A: Our algorithm is inspired by human cooperative pedagogy and our experiment did show its effectiveness in learning from human teachers. We agree that our algorithm is mainly for machine parameter learning, but when involving human teachers, algorithms inspired by human learning can be helpful.
>
> * Q: "the learning mechanism" What is the learning mechanism?
>     - A: Learning mechanism means the learning procedure followed by the learner, the parameter update rule like Eq. (1) or Eq. (6) to be more precise.
>
> * Q: " Yet, their findings cannot be directly applied to more practical teaching scenarios" Why? Please explain.
>     - A: The analysis in (Lessard et al. 2018) assumes the teacher (so-called omniscient teachers) knows the learner’s current parameter, $\nu^t$, and they share the same feature representation. These are much stronger assumptions than what ITAL needs and are too strong to be true in practical teaching scenarios. In addition, a concern of this setting is why the teacher still needs to teach with data when directly sharing the parameter with the student can be done. If the agents have different feature spaces, such a concern no longer exists. Moreover, if the teacher is human, knowing and analyzing a student's parameters directly becomes an overwhelming cognitive burden.
>
> * Q:  "Thus, we keep leveraging the greedy heuristic to model our cooperative teacher and generalize it to a non-omniscient teacher who doesn’t fully know the learner in every iteration" I do not understand the logic here.
>     - A: Sorry for not making it clear. We will clarify it here: as the teaching method in (Lessard et al. 2018) only applies to limited settings with omniscient teachers, we keep using the greedy teaching heuristic in our setting, where the teacher doesn’t necessarily know the learner’s status.
>
> * Q: "This can be done approximately if, in every round, the learner gives the inner products of ν t−1 and the data to the teacher as feedback." What does this mean intuitively?
>     - A: Using Eq. (2), the teacher needs to know the learner’s parameter at each iteration to calculate the second term. Without knowing the learner’s parameter, the teacher uses the lower bound of the second term to estimate it. Computation of the lower bound needs the inner product of the learner’s parameter and the data points. Discussion about the inner product can be found in the 2nd point above.
>
> * Q: " It has been shown that cooperative teachers using Eq. (5) can substantially " Do you mean equation 3?
>     - A: Sorry about this typo. It should indeed be Eq. (3). We’ll fix this in our final edition
>
> * Q: " to better approximate ν∗, in practice, we plug" What is the justification for this?
>     - A: The teacher-aware learner maximizes $\log q_{\nu}(x, y|\nu^{t-1})$ wrt. $\nu$ treating $\nu^{t-1}$ as constant. Namely, find a $\nu$ such that the probability of teaching student with $\nu^{t-1}$ using $x, y$ is maximized. In the learner’s mind, $\nu^{t-1}-\eta_t g_{x}(\nu^{t-1})$ is closest to $\nu^*$, so we calculate gradient of $\log q_{\nu}$ at $\nu = \nu^{t-1}-\eta_t g_{x}(\nu^{t-1})$ and conduct 1-step gradient descent.
>
> * Q: wording and expressions
>     - A: We’ll improve the wording and figures in the paper in the final version to address the related concerns.
> * Q: Ethical Concerns
>     - A: When we submitted this paper, the IRB was still under review. Now we have received the IRB approval.

---

> > ### Comment · Reviewer_WHuF · 2021-09-01
> > **Response to the response**
> >
> > I thank the authors for their responses to the questions that were raised by me and other reviewers. I do not find the first four replies to be all that helpful, unfortunately. When adopting and analyzing heuristics especially, details matter. Referring folks to other papers is not acceptable in my opinion. The fact that the paper (and the responses) build so directly and specifically on Liu et al is not desirable. I still do not understand the inner product idea from a logical perspective. The motivation seems entirely bound to the specifics of the models and the problem the authors (really Liu et al) propose. The comments about human learning are not acceptable. Inspiration is fine, but it doesn't rise to the level of matters that should be discussed in a paper. It is true that the experiment does include experiments with people that seem to work better than the chosen baselines. However, that this is true seems more accidental than a product of anything in the derivation of the approach. It is important to be very clear about these kinds of points. If the learning mechanism is an equation, it seems best to just refer to the equation?
> >
> > Although I was sympathetic to the paper in my initial review, I find myself wavering as a result of the responses.
> >
> > One last point: The IRB approval was obtained after submission of the paper? If so, this is a serious ethical violation.

---

> > > ### Author Response · Authors · 2021-09-02
> > > **Thank you for your feedback.**
> > >
> > > We apologize for the brevity and potential unclearness of our previous response to your first 4 questions. As first-time OpenReview users, we had unnecessary worries about the word limitation. Please allow us to provide more detailed responses to your questions (1-4) here.
> > >
> > > ---
> > > **Q1:** Details about teaching heuristic
> > >
> > > - **A1:** Our teacher uses greedy heuristics as in Eq (2). Every time the teacher selects the example that drives the student to the target parameter w^* the most, namely, the example that reduces the distance to the target parameter the most after one-step gradient descent. The first term in Eq. (2) measures the difficulty of an example. e.g. in the case of linear regression, this term is the squared error of an example. In the case of logistic regression, this term is the square of the misclassification probability. Usually, a more difficult example is also more informative and therefore brings larger parameter change to the learner.
> > > While the most difficult example can trigger the biggest change, it may not always be the most suitable one to use. The teacher also needs to make sure the change is in the right direction. To this end, the second term in Eq. (2) measures the correlation between the gradient of an example and the discrepancy between w^* and w^t. Hence, using the greedy heuristic, the teacher seeks an example that changes the student the most towards the correct parameter.
> > > The heuristic we are using has two advantages. First, it casts minimum assumptions to the teaching problem and can be applied to various learning tasks, including regression, classification, and inverse RL. Although other heuristics were proposed for specific teaching/learning scenarios, we believe our heuristic is more practical for settings without assumptions other than linear model and convex loss functions. Second, as the time complexity of finding the optimal teaching sequence is exponential to the rounds of interactions, and the parameter learning usually requires multiple rounds of communication, doing long-range planning easily becomes intractable to the teacher. The teaching heuristic in our paper allows the teacher to reply to the student in a prompt manner.
> > >
> > > ---
> > > **Q2:** Using the inner product as student feedback
> > >
> > > - **A2:** As our work is the first to propose teacher-aware learner for parameter learning, we start with learning the linear model. Thus, using inner products as student’s feedback messages is a natural choice, because the objective function of a linear model is a function defined on the inner product of the inputs and the model’s parameter. In addition, using inner products frees the choice of feature spaces. Like the kernel trick applied to SVM, using inner products as the message allows the agents to have feature spaces with different dimensions. The teacher doesn’t need to consider the learner’s exact feature representation, but only the output of their model, i.e., the inner products. Furthermore, given convex loss functions in our setting, the inner product provides additional optimization simplicity.
> > > 	Although, without additional specification of the learning tasks, we believe the inner product is a generic way of communication between the teacher and the student, by no means are we arguing that the inner product is the only way of communication from the student. We will consider other forms of feedback from the student, such as bandit feedback. In our future revision, we can compare the performance of using various forms of feedback.
> > > ---
> > > **Q3:** Connection with human learning
> > >
> > > - **A3:** Our setting is a multiagent learning problem involving a teacher and a learner, and the learner utilizes the helpfulness of the teacher to improve learning, both of which align with human pedagogy. In our paper, we didn’t claim that our algorithm models how humans learn because parameter learning is essentially on the machine side in human-machine interaction. What we are trying to demonstrate both theoretically and empirically is that when interacting with a human teacher, our algorithm (student learner) has a significant advantage over a naive learner.
> > > In our future work, we’ll investigate the human learning model with more comprehensive human study and compare the difference and similarities between the teacher-aware learner and human learner. We will be sure to make these points and our model’s relation with human pedagogy clearer in the final version.
> > > ---
> > > **Q4:** Clarification about Learning Mechanism
> > > - **A4:** Thank you very much for your advice, we will make it clear that the learning mechanism is referring to Eq. (1) in the revision.
> > > ---
> > > **Q5:** IRB issue
> > >
> > > - **A5:** We apologize for the confusion about our previous answer to this question. Our lab does have an institutional IRB approval covering human study with human-machine interactions for the grant supporting this work when we carried out the human study before the submission of this paper. The human study in this paper was designed and administered by researchers with the general IRB approval.
> > >
> > > ---
> > > We hope the above responses would address your concerns regarding Q1-4.

---

> > > > ### Comment · Reviewer_WHuF · 2021-09-05
> > > > **Responses are inadequate**
> > > >
> > > > I have reviewed the responses. They do not assuage my concerns. I have decremented my rating.

---

### Author Response · Authors · 2021-08-08
**Reply to Reviewers**

We would like to thank all reviewers for their time and detailed comments. We'll answer them in separate comment threads.

---

### Decision · Program_Chairs · 2021-09-27

**Decision:**

Accept (Poster)

**Comment:**

The paper studies a teacher-aware learning process in the context of iterative machine teaching. In particular, a new learner model is proposed that incorporates the teacher's intention into the likelihood function to learn faster.  Extensive experiments are performed on various tasks with synthetic and real data, including a user study on learning from human teachers. The reviewers acknowledged the importance of the studied problem setting and generally appreciated the results. However, the reviewers raised several concerns with the current version of the paper, including:  (a) the heuristic nature of the teaching model is not analyzed; (b) the learner feedback as an inner product is not well motivated or explained.; (c) the theoretical guarantees only show that under certain assumptions the proposal cannot do worse than a naive learner; (d) the learning mechanism referred to in the paper is not well-defined; (e) some of the important details of the experimental set-up for the online IRL experiments and the study with human teachers are missing from the main paper. I want to thank the authors for their detailed responses that helped in answering some of the reviewers' questions. While the overall assessment is positive, the paper still stands as borderline. Nevertheless, this is interesting and potentially impactful work. The reviewers have provided detailed and constructive feedback to the authors. We strongly encourage the authors to incorporate the reviewers' feedback when preparing a revision of the paper.